# The prefrontal cortex encodes task-identity information and flexibly adjusts its sensory processes as a function of the specific ongoing task

Axel Mouille, Corentin Gaillard◉, Elaine Astrand◉¤a, Claire Wardak¤b, Julian Luis Amengual‡¤c*, Suliann Ben Hamed◉‡*

Institut des Sciences Cognitives Marc Jeannerod, CNRS UMR 5229, Université Claude Bernard Lyon I, Bron, France

◉ These authors share second-authorship of the manuscript.
‡ These authors share last authorship on this work and are corresponding authors.
¤a Current address: School of Innovation, Design, and Engineering, Mälardalen University, IDT, Västerås, Sweden
¤b Current address: Université de Tours, INSERM, Imaging Brain & Neuropsychiatry iBraiN U1253, Tours, France
¤c Current address: Instituto de Investigación Sanitaria La Fe, Plataforma de Big Data, IA y Bioestadística, Valencia, Spain
* benhamed@isc.cnrs.fr (SBH); julian.amengual@gmail.com (JLA)

## Abstract

The prefrontal cortex (PFC) plays a key role in selecting, maintaining, and representing sensory information, and in its integration with our current internal goals and expectations, implementing such cognitive processes as executive functions, attention, decision-making, and working memory. Performing computation over all of these functional cognitive processes and dynamically shifting from one to the other based on task demands requires a complex functional organization and a high degree of coding flexibility. The PFC cells show a non-linear mixed selectivity characterized by specific tuning for multiple task- and behavior-related parameters. This non-linear mixed selectivity is thought to allow for a high-dimensional representation of information. Here, we asked if the PFC is mainly involved in the specific task-parameters representation, or if it additionally holds a higher-order representation for task-identity. We thus trained two macaques to perform three different tasks: A memory guided saccade task and two detection tasks involving different attention mechanisms. Multi-unit activity was recorded in the frontal eye field, bilaterally, while the monkeys performed these three tasks in the same session. Using demixed Principal Component Analysis, we found a two-dimensional neural state that characterized each of these tasks. The lower dimensional representations of the activity recorded during the performance of the two attentional tasks were more similar to each other than to the memory-guided saccade task. Furthermore, we report that task and spatial information are non-linearly mixed, a signature of a high-dimensional neural representation.

**Data availability statement:** All relevant data and scripts can be downloaded from the following OSF link: https://osf.io/z8eh9/.

**Funding:** This work was supported by the Centre National de la Recherche Scientifique (CNRS, http://www.cnrs.fr) to S.B.H., Direction Général de l'Armement (DGA, https://www.defense.gouv.fr/dga) to E.A. and S.B.H., Fondation pour la Recherche Médicale (FRM, http://www.frm.org) to E.A. and S.B.H., Fondation de France (FDF, 00089976 http://www.fondationdefrance.org) Berthe Fouassier to A.M and S.B.H., and french National Research Agency (ANR, http://anr.fr) to S.B.H. (ANR-11-BSV4-0011, ANR-11-LABX-0042, ANR-IDEX-0007). The funders had no role in study design, data collection and analysis, decision to publish, or preparation of the manuscript.

**Competing interests:** The authors have declared that no competing interests exist.

**Abbreviations:** cBCI, cognitive brain-computer interface; CTOA, cue to target interval; dPCA, demixed principal component analysis; FEF, frontal eye field; LIP, lateral intra-parietal area; MUA, multi-unit activity; PCA, principal component analysis; PFC, prefrontal cortex; RNN, recurrent neural network.

Overall, this indicates that PFC encodes task-identity information and flexibly adjusts its sensory processes as a function of the specific ongoing task.

## Introduction

The prefrontal cortex (PFC) produces internal representations of our actions and of our environment. Its computations contribute to the selection, maintenance, and filtering of sensory information, in order to integrate the representation of the current task parameters with overarching internal goals and expectations. As a result, the PFC functionally contributes to several cognitive processes that include executive functions [1–3], attention [4–7], decision-making [8,9] and working memory [10,11]. An important question is how the PFC implements these different functions both at the single cell and at the population level. Electrophysiological studies have shown that the PFC displays cells tuned to multiple task- and behavior-related parameters, a property called non-linear mixed selectivity, which provides computational advantages for information transmission to downstream neurons [12,13]. In particular, mixed-selectivity allows a given neuronal population to code multiple variables (or cognitive functions) with minimal interference from one variable on the readout of the other variables. Indeed, non-linear mixed-selectivity has been proposed to result in a high-dimensional representation of multiple sensory information patterns of stimuli (such as position, color or shape) [12]. Such an encoding strategy allows for the simple readout of complex patterns of behavior, using for example linear classifiers. In the present study, we aim to describe whether and how this mixed-selectivity contributes to the coding of task-identity in the PFC, and how it articulates task-identity coding with that of spatial information and possibly facilitates the flexible adjustment of the representation of spatial factors to specific task demands.

While the ability of the PFC—and the FEF in particular—to encode multiple variables and task contingencies is well-established (e.g., visual and motor [14–16], working memory [17,18] and spatial attention [19,20] signals), a key unresolved question concerns how such flexible representations are organized to support generalizable, adaptive behavior across tasks. Recent theoretical frameworks propose that flexible cognition may rely on compositional neural codes, in which complex tasks are constructed from modular, reusable components of computation [21,22]. Recent computational work proposes that recurrent neural networks (RNNs), such as those described in prefrontal cortical circuits, utilize shared dynamical motifs—recurring patterns of neural activity implementing specific computations through dynamics such as attractors, decision boundaries, and rotations—to encode task-related information and support flexible multitasking [23,24]. Yet, direct neurophysiological evidence for such compositionality in prefrontal areas remains scarce [25,26]. In this study, we address this gap by investigating whether the FEF not only encodes individual task parameters, but also supports higher-order task identity representations that reflect the compositional structure of cognitive operations. By applying demixed principal component analysis to multi-task electrophysiological data in monkeys, we uncover

low-dimensional neural states that capture both the identity and proximity of tasks, supporting the notion that the FEF may contribute to cognitive flexibility by encoding abstract, compositional task structures rather than mere sensorimotor mappings.

To address this question, we trained two macaque monkeys to perform three different tasks in the same recording sessions: (1) a centrally cued manual target detection task, the target being presented in the periphery of the visual field; (2) a cued manual target detection task, in which the cue is presented at the expected location of the target in the periphery of the visual field; (3) a memory-guided saccade task in which an eye movement has to be performed, upon instruction, towards a previously cued location. Each of these tasks recruited specific cognitive functions: Endogenous attention, exogenous attention, and spatial working memory. While all tasks involved spatial working memory processes, they differed in their attentional demands and responses. The attentional tasks included specific landmark presentations during the delay and required a manual response to the target, whereas the spatial working memory task required a saccadic response to the target. As monkeys were performing these tasks in independent blocks of trials, recording probes were used to record multi-unit activity (MUA) from both frontal eye fields (FEF), a region in the prefrontal cortex linked with top–down attentional processes [7,27–29] and working memory [18,30]. Using demixed principal component analysis (dPCA [31,32]), we extracted latent components associated specifically with either sensory information or task identity. We found that the neuronal processes underlying each task were encoded in two different orthogonal components, one component reflecting the encoding of either attention versus working memory, and the other component reflecting the encoding of endogenous versus exogenous attentional enhancement. This possibly suggests that task-identity is represented at the same hierarchical level as specific task parameters such as position. However, we also identified an additional component that encoded the interaction between task-identity and sensory processing, indicating that these two sources of information (task and position of the sensory information) could be decoded simultaneously by a single linear readout, though the readout of sensory information varied across tasks. This result suggests that, although the FEF encodes some aspects of task identity and spatial information independently, some other aspects of task identity influence how spatial information is coded, such that the specific state of one variable influences the specific computations implemented by the other variable.

All in all, these results suggest that the prefrontal cortex encodes sensory information and task-identity in a high-dimensional neural representation. In addition, some aspects of spatial information are represented independently from task-related information, while other aspects of task-related information influence spatial information coding, suggesting that task-identity is represented at the same hierarchical level as specific task parameters such as position. At the fundamental level, it will be important to continue this work in order to better understand the specific dimensions that the prefrontal cortex uses to represent task identity. At the translational level, this work opens new venues for the implementation of brain computer interface technologies based on machine learning algorithms able to extract complex cognitive information from real-time PFC recordings.

## Methods

### Surgical procedure and FEF mapping

Two male rhesus monkeys (*Macaca mulatta*) weighing between 6 and 8 kg underwent a single surgery during which they were implanted with two MRI compatible PEEK recording chambers placed over the left and the right FEF hemispheres respectively, as well as a head fixation post. Gas anesthesia was carried out using Vet-Flurane, 0.5%–2% (Isofluranum 100% at 1,000 mg/g) following an induction with Zolétil 100 (Tiletamine at 50 mg/ml, 15 mg/kg and Zolazepam, at 50 mg/ml, 15 mg/kg). Post-surgery pain was controlled with a morphine painkiller (Buprecare, buprenorphine at 0.3 mg/ml, 0.01 mg/kg), 3 injections at 6 hr-intervals (first injection at the beginning of the surgery) and a full antibiotic coverage was provided with Baytril 5% (a long-acting large spectrum antibiotic, Enrofloxacin 0.5 mg/ml) at 2.5 mg/kg, one injection during the surgery and thereafter one each day during 10 days. A 0.6 mm isotropic anatomical MRI scan was acquired

post surgically on a 1.5T Siemens Sonata MRI scanner, while a high-contrast oil-filled 1 mm × 1 mm grid was placed in each recording chamber, in the same orientation as the final recording grid. This allowed a precise localization of the arcuate sulcus and surrounding gray matter underneath each of the recording chambers. The FEF was defined as the anterior bank of the arcuate sulcus and we specifically targeted those sites in which a significant visual, premotor and/or oculomotor activity was observed during a memory guided saccade task at 10–15° of eccentricity from the fixation point. In order to maximize task-related neuronal information at each of the 24-contacts of the recording probe, we only recorded from sites with task-related activity observed continuously over at least 3 mm of depth. All surgical and experimental procedures were approved by the local animal care committee (C2EA42-13-02-0401-01) approved by the French Ministry of Research and in compliance with the European Community Council, Directive 2010/63/UE on Animal Care.

### Behavioral tasks and experimental setup

In all behavioral tasks, monkeys were seated in front of a computer screen (1920 × 1200 pixels and a refresh rate of 60 Hz) at a distance of 45 cm with their head fixed. To initiate a trial, the monkeys had to hold a bar in front of the animal chair, thus interrupting an infrared beam. They were required to hold the bar throughout the trial until a target stimulus was presented, otherwise the trial was aborted. The eccentricity of the visual stimuli was adjusted from day to day between 10–15°, to match the preferred spatial location of the multi-unit activity recorded on the 48 contacts of both recording probes.

**Exogenous 100% validity cued luminance change detection task (*Exo*).** The task is a 100% validity cued luminance change detection task with temporal distractors (Fig 1A). To initiate a trial, the monkeys had to hold a bar in front of the animal chair, thus interrupting an infrared beam. A blue fixation cross (size: 0.7 × 0.7°) appeared in the center of the screen. Monkeys were required to hold fixation throughout the entire trial, within a fixation window of size 4° × 4°. Failing to do so aborted the trial and another trial started again. Four gray landmarks (0.5 × 0.5° for monkey M1, 0.68 × 0.68° for monkey M2) were presented simultaneously and equidistantly with the fixation cross. These landmarks were located, in the upper right, upper left, lower left and lower right quadrants of the screen, thus defining the corners of an illusory square. Their specific eccentricity was adjusted from day to day between 10–15°, to match the preferred spatial location of the multi-unit activity recorded on the 48 contacts of both recording probes. After a variable delay from fixation onset, ranging between 700 and 1900 ms, a green square was presented for 350 ms, indicating to the monkey in which of the four landmarks the rewarding target change in luminosity would appear. This green square (from now on, the cue) was presented on top of the peripheral landmark to be attended to. After the cue presentation, the monkeys needed to orient their attention to the target landmark in order to monitor it for an increase in luminosity while maintaining eye fixation onto the central blue cross. This increase in luminosity (from now on, the target) could occur anywhere between 500 and 2,800 ms from cue onset. The increase in luminosity was titrated during preliminary behavioral experiments, independently for each of the four possible target positions, such that behavioral performance during the Exo task would be in the range of 85–90% at each location. In order to receive their water or juice reward, the monkeys were required to release the bar (thus restoring the infra-red beam) in a time window of 200–700 ms following the target onset. This event accounted as a hit trial, while failing to respond to the target resulted in miss trials. In order to assure that the monkeys were correctly orienting their attention towards the cued landmark, unpredictable changes in the luminosity identical to the awaited target luminosity change could take place at the uncued landmarks (from now on, distractors). On each trial, from none to three such unpredictable distractors could take place, no more than one per uncued landmark position. Monkeys were trained to avoid responses to these distractors as a response to a distractor interrupted the trial and was counted as a false alarm trial.

**Endogenous 100% validity cued luminance change detection task (*Endo*).** The endogenous task was identical to the exogenous task in all aspects, including parameters of target luminosity increase, except for the location and size of the cue (Fig 1B, panel 1). The cue stimulus consisted of a green square (0.2 × 0.2° for monkey M1 and 0.3 × 0.3° for

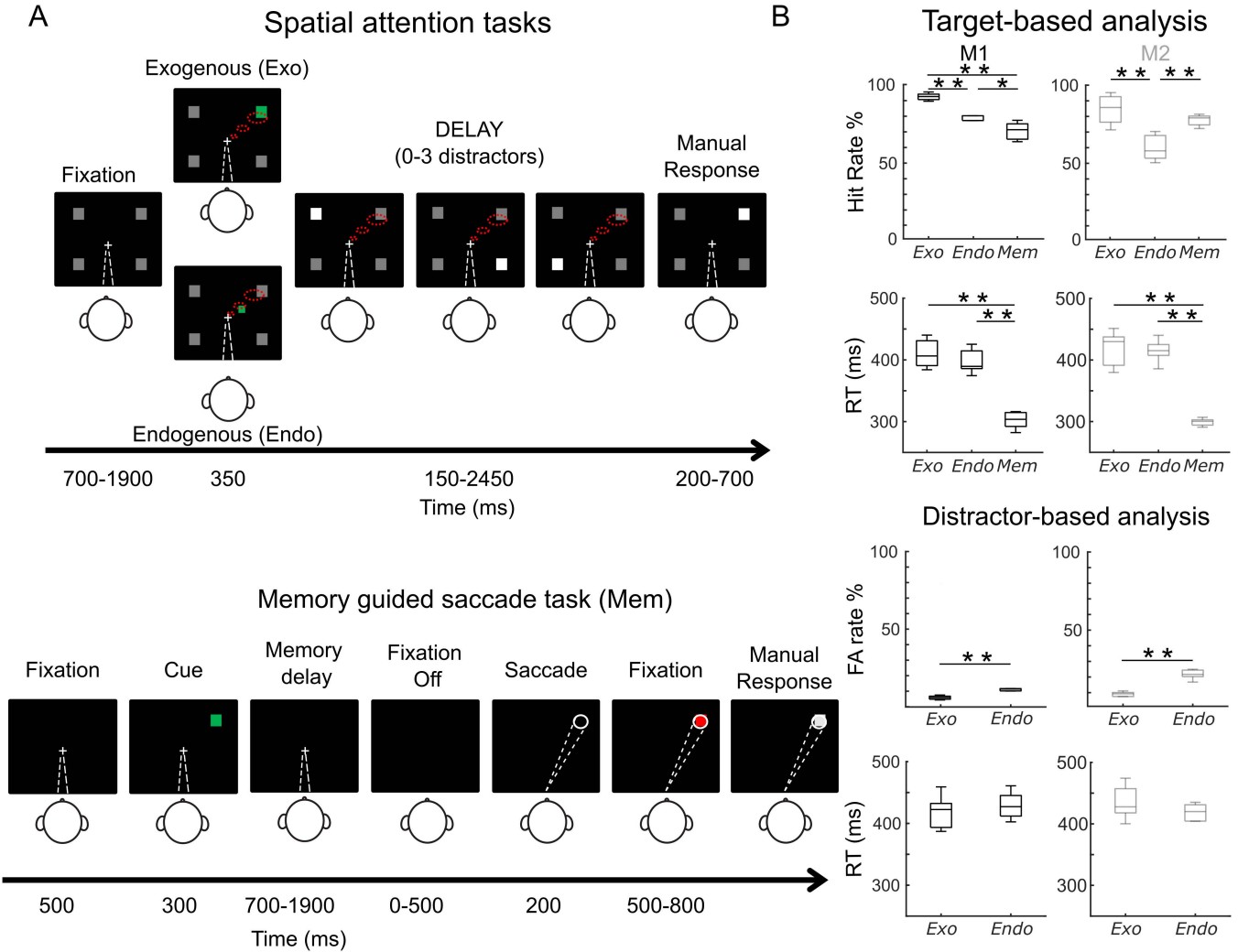

**Fig 1. Description of the tasks and main behavioral results. (A)** Monkeys performed three behavioral tasks: The endogenous (*Endo*) and exogenous (*Exo*) attention tasks consisted of 100% validity cued target-detection task with distractors. To initiate the trial, monkeys had to hold a bar with the hand and fixate their gaze (white dashed line, first screen) on a central cross on the screen during the whole trial. A cue (green square, second screen) was presented indicating to the monkey in which of the four landmarks the target change in luminosity would appear. Monkeys received a liquid reward for releasing the bar 200–700 ms after target presentation onset. Monkeys had to ignore any uncued event (distractors). Importantly, they had to detect the stimuli using selective spatial attention (red dashed lines). In the memory-guided saccade task (Mem), a peripheral stimulus appeared 300 ms following fixation (500 ms). The monkeys had to hold the position in memory, throughout a variable delay of 700 to 1900 ms, until the fixation cross disappeared. At the extinction of the fixation cross, the monkeys had 500 ms to execute a saccade to the previously presented stimulus location. At the end of the saccade, the peripheral stimulus reappeared and the monkeys had to wait and respond when a target stimulus was presented. **(B)** Behavioral performance. On the top part, boxplots represent the distributions of hit rates and reaction times (RT) over all sessions (*n* = 12). On the bottom part, boxplots show distributions of the false alarm rate (FA rate) and reaction time (RT) of responses to distractors over all sessions (*n* = 12, non-parametric Wilcoxon signed rank test *$p$ < 0.05, **$p$ < 0.01). Data and code to Fig 1B can be found at https://osf.io/z8eh9/.

monkey M2) and it was presented close to the fixation cross in the same direction as the landmark to be attended (at 0.3° for monkey M1 and at 1.1° for monkey M2, relative to the fixation point).

**Memory-guided saccade task (*Mem Sacc*).** A standard oculomotor memory task was used (see Fig 1B, panel 3) in which the monkeys needed to fixate a blue cross in the center of the screen (0.3 × 0.3°) in order for the trial to be initiated.

Fixation within a window of 4 × 4° around the fixation cross was required, otherwise the trial was aborted. After 500 ms of fixation, a peripheral stimulus (size: 0.3 × 0.3°) appeared for 300 ms. The monkeys had to hold the position in memory, throughout a variable delay of 700–1900 ms, until the fixation cross disappeared. At the extinction of the fixation cross, the monkeys were required to execute a saccade to the previously presented peripheral stimulus, within a window of 8 × 8° around it. The monkeys had 500 ms to execute the saccade to the correct position or the trial was aborted. After a correct saccade, fixation on the empty screen had to be maintained 200 ms before the peripheral stimulus reappeared. The monkeys then had to maintain fixation on this peripheral stimulus within a window of 4 × 4° and wait until the red color changed to gray (variable delay of 500–800 ms). At color change, the monkeys had to release the bar (thus restoring the infrared beam) within a time window of 150–800 ms for a liquid reward.

Within any recording session (M1: $n=7$; M2, $n=5$), the monkeys performed the three tasks consecutively. The order of the attentional tasks was changed randomly between sessions, while the memory-guided saccade task was always presented last, as monkeys displayed a strong behavioral preference for this task -though with a low success rate due to not sufficiently precise saccades- and refused to work on the attentional tasks when these were presented after the memory guided saccade task.

## Neural recordings

Simultaneous recordings in the two FEFs were carried out using two 24-contact Plexon U-probes. The contacts had an interspacing distance of 250 µm. Neural data was acquired with PlexonOmniplex neuronal data acquisition system (Omniplex). The data was amplified by a factor of 400 and digitized at 40,000 Hz. The neuronal data was low-cut filtered at 300 Hz. All analyses were performed on the MUA recorded from each of the 48 recording contacts (M1: number of sessions = 7, total number of channels = 336; M2: number of sessions = 5, total number of channels = 240). A threshold was applied independently for each recording contact and before the actual task-related recordings started, such that the crossing of this threshold resulted in multiunit spike trains. This threshold corresponded to 3 standard deviations from mean activity during unconstrained eye movements by the monkeys while exploring a blank screen. This threshold was confirmed during the offline analyses to optimize single unit activity extraction for each channel. This readjustment impacted MUA final thresholding values. Spike sorting was performed, on a subsample of channels (M1: $n=94$; M2: $n=69$), using the Offline Sorter (Plexon), on the broadband time signal (bandpass filtered between 300 and 3,000 Hz) of the entire session prior to assigning the signal to the different tasks. Only spikes that could be unambiguously assigned to a same neuron throughout all the tasks were used for the SUA-based analyses (M1: $n=43$; M2: $n=15$). The "Clusters vs. Time" view was used (using the most informative principal component pairs) to check how the waveform changed as a function of time throughout the session. Only units that could be continuously tracked through the entire session, with no drift or a smooth continuous drift in PCA cluster attributes were used for the SUA-based analyses. Examples of spike sorted waveforms tracked in the three different tasks are presented in S1 Fig. All further analyses of the data were performed in Matlab (The Mathworks, Natick, Massachussetts).

## Data pre-processing

In each task, MUA/SUA activity was epoched per trial. Spike trains were filtered with a Gaussian kernel ($\delta=60$ ms). First, MUA/SUA activity previous to the cue onset was epoched from −200 ms to 0 ms with respect to the onset of the Cue for the Pre-Cue analysis. Then, MUA/SUA activity was epoched between 500 ms to 0 ms prior to Target onset for the Delay analysis. To extract the averaged firing rates for dPCA analysis, these epoched MUA/SUA data were baseline corrected (independently for each channel and each trial) relative to the time period from −500 ms to −200 ms before the cue. This reduced trial-to-trial variability in baseline activity levels and reduced individual trial differences.

## MUA selectivity

Spatially selective channels during the delay epoch were identified as follows. First, a non-parametric Wilcoxon test was used to identify responsive channels, i.e., channels for which the activities observed during the baseline and during the post-cue period (800–900 ms following cue onset, only considering trials for which target or fixation off event happened 1,000 ms after cue presentation) were significantly different for at least one of the four possible spatial cued positions ($p < 0.05$). For each responsive channel, a non-parametric two-way ANOVA Task × Position (David Stern, https://www.mathworks.com/matlabcentral/fileexchange/44308-randanova2, this method uses the Still and White's [33] approach to calculate $P$-value for interaction effect) was then implemented on the average activity in the post-cue period in order to identify the Task related channels, the spatially selective channels as well as the mixed selectivity channels ($p < 0.01$). All $p$ values were corrected for multiple comparisons using a Holm-Bonferroni correction. Note that the proportion of selective cells did not change substantially across different analysis time windows (S5 Fig).

## Demixed PCA analysis

Recent studies have shown that neurons in the prefrontal cortex show simultaneous significant tuning to multiple task-related parameters, a property called mixed selectivity [12]. This feature is a hallmark of high dimensionality of the neuronal population. Due to this property, the activity structure of the population can be estimated by applying a dimensionality reduction method to the recorded activity such as Principal Component Analysis (PCA). Using this method, it is possible to extract a number of latent variables (principal components) that capture independent sources of data variance providing a description of the statistical features of interest [34]. However, the PCA method is blind to the sources of variability of the data and does not take into account the task related parameters, mixing these sources of information within each of the extracted latent variables [32]. Here, we wanted to study and compare the population activity structure in the three different tasks and describe how much the variance in the neuronal population can be explained by task-related aspects such as task, spatial target location, and their interaction. To address this question, we applied dPCA (demixed principal component analysis [31,32]) on the data which captures the maximum amount of variance explained by predefined sources of variability in each extracted component and reconstruct the time course of the category-specific response modulation. A recent publication indicates that dimensionality reduction methods are accurate in the absence of spike sorting [35]. In other words, neural dynamics and scientific conclusions are comparable when using multiunit threshold crossing or sorted neurons when using dimensionality reduction methods. This is particularly relevant to the present study. Indeed, 10–15 dimensions capture over 90% of the variability of trial-averaged neural population activity when monkeys perform the different tasks. This is associated with low error in manifold estimation [36]. As a result, due to the theory of random projections, neural population dynamics can be accurately estimated from our recordings using MUA instead of SUA, since the former is a low-dimensional representation of the latter. In the present work, the dPCA is applied on MUA activities.

Demixed PCA was applied in the two different time intervals (*Pre-Cue* and *Delay*). The purpose of the first dPCA analysis was to identify specific task-related components during the Pre-Cue period. Then, another dPCA was performed during the delay period, with the aim to extract task and spatial-related components, as well as components associated with their interaction (non-linear mixed selectivity). Both analyses were performed using appropriate Matlab based toolbox developed by Kobak and colleagues [32]. We considered the 20 first demixed components that accounted for >90% of the total variance.

We used the decoding axis of each dPC (assigned to task in the *Pre-Cue* analysis, or task, position and interaction in the *Delay* analysis) as a linear classifier to decode the different types of trials (*Endo*, *Exo,* and *Mem Sacc* for task, and *Up Left*, *Up Right*, *Down Left,* and *Down Right* for position), following the procedure described by Kobak and colleagues [32]. This method allows to quantify the capacity of each demixed component to classify a trial between the classes of a given category. To extract the statistical significance of this accuracy, we shuffled 100 times all available trials between classes

and we thereby computed the distribution of classification accuracies expected by chance. For each session, the chance-level was considered as the maximal accuracy value obtained across all randomizations.

## Results

### Monkey performance is task-dependent

The present study aims to describe and compare, in the PFC, the electrophysiological underpinnings of spatial information encoding during the performance of different tasks. To this end, we recorded MUA from bilateral monkey frontal eye field (FEF), an area involved in the control of covert spatial attention [5,7,37,38] and saccades [16,39–41], of two macaque rhesus monkeys while they performed three different tasks: An endogenous attention task (*Endo*), an exogenous attention task (*Exo*) and a memory-guided saccade task (*Mem*). We used two probes (one in each recording site) containing 24 contacts each in both monkeys, which were implanted in each recording session (number of sessions: M1, $n=7$; M2, $n=5$). The median number of trials per task was as follows: Exogenous task, M1: 318 (iqr = 59); M2: 380 (iqr = 62); Endogenous task, M1: 357 (iqr = 67); M2: 589 (iqr = 171); Memory guided saccade task, M1: 244 (iqr = 39); M2: 302 (iqr = 184). These three tasks were presented consecutively in the same session in a block design. The schematic of these tasks was very similar (see Fig 1A and "Methods" section for an extensive description of these tasks): At the beginning of each trial, a visual cue was presented in one of the four possible screen quadrants, and after a variable and randomized time interval monkeys had to produce a response. However, the actual task differed, and the monkeys needed to adapt their cognitive processes to perform each task. The exogenous and endogenous attentional tasks consisted of a 100% validity cued luminance change detection task in which monkeys had to respond manually to a target located peripherally in the cued quadrant. In the exogenous version, the cue was shown in the peripheral landmark to be attended to (thus, in the same position as the target). In contrast, in the endogenous attention task the cue was presented close to the fixation cross pointing towards the landmark to be attended. In both endogenous and exogenous attention tasks, distractor stimuli were presented during the cue to target interval (CTOA), and monkeys needed to ignore them (otherwise, the trial was cancelled). These two tasks were used to recruit attentional processes [29]. In the memory-guided saccade task, monkeys were required to hold the position of a spatial cue in memory during a variable amount of time and, then, to perform a saccade towards that memorized spatial location on the presentation of a go signal. Importantly, and in contrast with the attentional tasks, monkeys were required to perform a spatially oriented oculomotor response rather than a manual non-oriented response. This task thus recruited spatial working memory processes [29,16,41]. Although these three tasks showed some similarities in their structure, they recruited different cognitive processes. While monkeys needed to enhance visual attention in the cued quadrant to respond to the target onset in the exogenous and endogenous attentional tasks [29,42,43], they needed to memorize the location of the cue to produce an oriented eye movement towards that location in the memory-guided saccade task. Sensory conditions were also different. Indeed, in the attentional tasks, peripheral landmarks were present all throughout the trial, while they were absent in the memory guided saccade task. While this probably did not impact pre-cue neuronal activity, it possibly impacted post-cue task differences observed in the activity.

Both monkeys showed different behavioral patterns as a function of the type of task they performed. Non-parametric Friedman test (Tasks as repeated measures, $df=2$, performed across sessions) revealed that hit rates (M1: $\chi^2(5) = 12.29$, $p=0.002$; M2: $\chi^2(5) = 7.6$, $p=0.0224$) and reaction times (RT) (M1: $\chi^2(5) = 12.29$, $p=0.0021$; M2: $\chi^2(5) = 7.6$, $p=0.0224$) were different across tasks (Fig 1B). This reflects different task demands.

Both monkeys were able to discriminate the change in luminosity of the cued landmark while ignoring distractors in the spatial attention tasks, though their performance differed depending on the type of task. Indeed, both monkeys showed a higher hit rate when performing *Exo* than *Endo*, (Fig 1B M1: 92% versus 78%, Wilcoxon ranksum test, $Z=3.07$, $p=5.8e-04$; M2: 87% versus 60%, Wilcoxon ranksum test, $Z=2.09$, $p=0.0079$). M1 showed a lower

hit rate in memory-guided saccade task than in *Exo* and *Endo*, while M2 showed significantly lower hit rate in *Endo* than in *Mem* (Fig 1B M1: 72% versus 92%, Wilcoxon ranksum test, $Z = 3.07$, $p = 5.8e − 04$, 72% vs.78%, $Z = 2.17$, $p = 0.0233$; M2: 60% versus 78%, $Z = 2.51$, $p = 0.0079$). Regarding the RT, and expectedly, both monkeys had slower manual response times in attentional tasks than saccadic response times in the *Mem* task (M1, Friedman test, $\chi^2(5) = 12.29$, $p = 0.0021$; M2, Friedman test, $\chi^2(5) = 7.6$, $p = 0.022$. M1: *Exo* versus *Mem*, Wilcoxon ranksum test, $Z = 3.07$, $p = 5.8e − 4$, *Endo versus Mem*, Wilcoxon ranksum test, $Z = 3.07$, $p = 5.8e − 4$; M2, *Exo versus Mem*, Wilcoxon ranksum test, $Z = 2.51$, $p = 0.0079$; *Endo versus Mem*, $Z = 2.51$, $p = 0.0079$). Regarding the attentional tasks, although no significant difference was found, RTs tended to be slower for the Endo than for the *Exo task* as reported in other studies [29].

False alarm rates (measured as the ratio between the number of responses to distractor and the total number of distractors shown) and reaction time in false alarms were analyzed only for the attentional tasks, since no distractors were used in *Mem*. Both monkeys showed lower false alarm rates in Exo than in Endo (Fig 1B M1: 7% versus 13%, Wilcoxon ranksum test, M1: $Z = 2.56$, $p = 0.007$; M2: 10% versus 21%, Wilcoxon ranksum test, $Z = 0.63$, $p = 0.0079$). Monkeys did not show differences in reaction times in the false alarms between the two attentional tasks. Notably, responses were slower in false alarms than in hits for M1 in *Endo* (Wilcoxon ranksum test, $Z = 1,79$, $p = 0.0175$).

## Spiking activity in the FEF encodes task- and sensory- related information simultaneously

We examined the FEF neural responses during the performance of these three tasks. To this end, we averaged the smoothed firing rates in the time interval −500 ms to 1,000 ms locked to the cue for each of the three tasks. As reported in previous studies [7,42,44,45], these recorded neurons showed enhanced responses when the cue was presented in or was cueing towards their preferred spatial position, relative to when it was presented in the least preferred spatial location, and this was true for all three tasks (Fig 2A and 2B, see S2 Fig for SUA analysis). Similar results were found when data were locked to the target onset (S3 Fig). Spatial selectivity however varied between the three tasks. Indeed, we found a significant task-effect, across the entire population, on the response to the preferred position (non-parametric Friedman test, performed across MUAs, with tasks as repeated measures, $\chi^2(2) = 39.3$, $p = 2.89e − 09$).

Specifically, we observed that some cells were recruited during the performance of the three tasks during the delay period (Fig 2A, gray box). We measured the proportion of cells that expressed a significant spatial selectivity during the delay period of the three tasks. Results of this measure are represented in Fig 2C as a Venn diagram. Interestingly, we found that 17% of cells were tuned to position in all three tasks, indicating that these cells encoded spatial information both during spatial attention and spatial working memory. In addition, we found neurons that showed mixed selectivity for both task and spatial information, like the one represented in this figure. Therefore, we classified all recorded cells based on either their selectivity to position, task, or to simultaneous position and task selectivity. Based on the statistical significance of the firing rate modulation for these two parameters (spatial selectivity and task selectivity), we found that 32% of cells were uniquely selective for the task, 6% were uniquely selective for position, and 60% showed mixed selectivity for both parameters: 38% showed no interaction between task and position (corresponding to linear mixed-selectivity) and 22% showed an interaction between task and position (corresponding to non-linear mixed-selectivity – Fig 2D). S2 Fig duplicates the analyses presented in Fig 2 for SUAs. S4 Fig represents an example and the population activity for each category of MUAs/SUAs. This categorization was robust against possible sources of noise in the data (S1 File Supporting information: A, MUA; B: SUA). For MUA, selectivity did not depend on the post-cue analysis window (MUA: S5A Fig), and only marginally did so for SUA (SUA: S5B Fig). As expected, no position-related nor mixed-selectivity MUAs or SUAs were identified during the pre-cue epoch (−200 to 0 ms prior to cue presentation, S5 Fig). These results indicate that the FEF encodes spatial information and the underlying cognitive processes recruited by each task through mixed-selectivity neurons. Importantly, we found that 94.6% of channels that were exclusively task selective during the delay, were also task-selective in the pre-cue period.

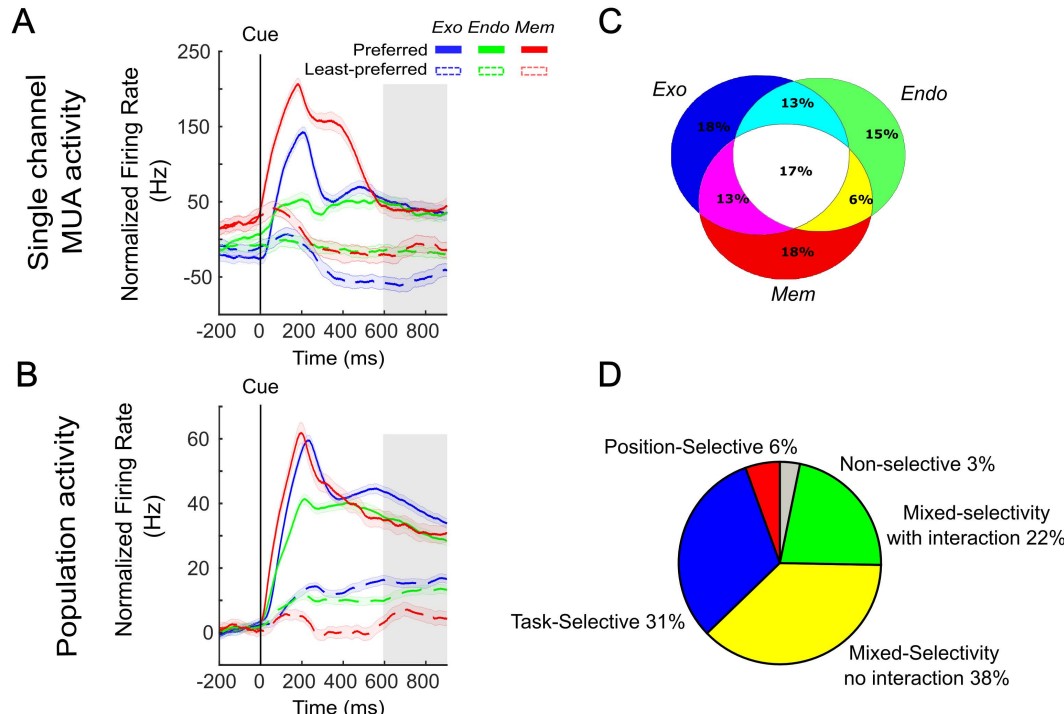

**Fig 2. Mixed selectivity corresponds to the ability of a neuronal population to encode simultaneously independent sources of information: (A) Firing rate of a representative MUA aligned on the cue (200 ms before the cue onset to 1,000 ms after the cue onset) in exogenous task (blue), endogenous task (green) and memory guided saccade task (red) for preferred (solid line) and non-preferred (dash line) position.** The data were baseline corrected (−500 to −200 ms before cue onset) on individual trials and channel. The gray box corresponds to the delay period during which neuronal activities were analyzed for subplots C and D. **(B)** Mean firing rate of the MUA neuronal population in the three tasks aligned on cue presentation (200 ms before the cue onset to 1,000 ms after the cue onset) in the exogenous task (blue), the endogenous task (green) and the memory guided saccade task (red) for preferred (full line) and non-preferred (dash line) position. The data were baseline corrected (−500 to −200 ms before cue onset) on individual trial and channel. Gray box corresponds to the delay period during which neuronal activities were analyzed for subplots C and D. **(C)** Venn diagram representing the proportion of cells showing a spatial tuning in each of the three tasks. **(D)** Pie chart corresponding to the proportion of neurons tuned to only position (red), task (blue) and both (Additive mixed-selectivity; yellow; Mixed-selectivity with interaction; green). Non-selective neurons are plotted in gray. Wilcoxon ranksum test was performed between baseline (−100 to 0 before the cue onset) and time interval (800 to 900 ms after the cue onset) for each channel and position across trials. A non-parametric 2-Way-ANOVA was performed on selective channels with Bonferroni–Holm correction on spatial and task factors on time interval from 600 to 900 ms after the cue onset ($p<0.01$). Data and code to Fig 2 can be found at https://osf.io/z8eh9/.

Fig 3 aims to characterize the impact of mixed-selectivity, with and without interactions, on attention selectivity spatial indicators prior to target presentation. Some MUAs had a firing rate that was significantly higher for a well-identified preferred position compared to the three other positions in at least one of the three tasks. Such a spatially organized response pattern could be related to the match between our task configuration and the MUAs' receptive field location, where one of the target positions coincided with the center of the receptive field. However, such a response pattern was predominantly identified in the Exo task (Mixed-selectivity without interaction: Exo: 18%, Endo: 5%, Sacc: 5%; Mixed-selectivity with interaction, Exo: 46%, Endo: 24%, Sacc: 18%). This indicates that, independently of the relative match between task configuration and MUA receptive field, the response pattern changed between the Exo task and the other tasks and became spatially less resolved. This is exemplified in Fig 3, which shows representative single MUAs (Fig 3A, mixed-selective MUA finely tuned receptive field; 3B, mixed selectivity MUA with broad receptive field; MUA and SUA examples showing finely-tuned receptive fields or broad receptive fields are presented in S6 Fig) and population-level responses (Fig 3C, average population mixed-selective MUA without interaction; 3D, with interaction). To further analyze

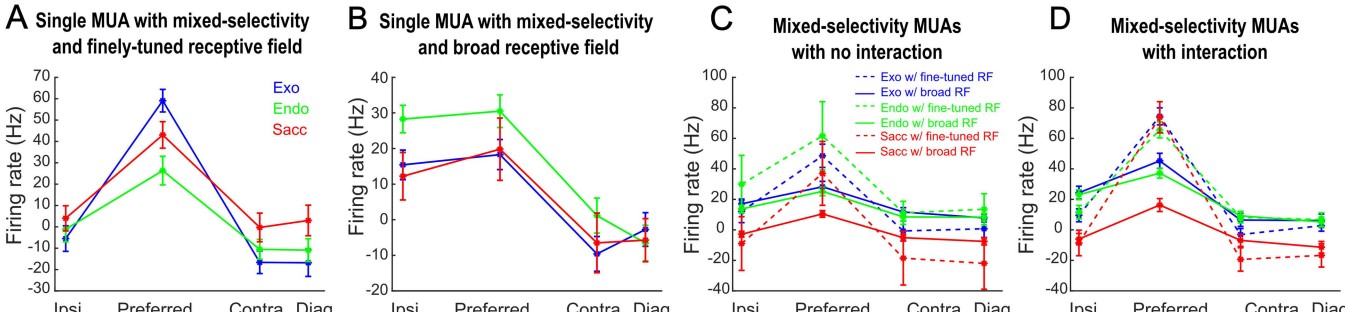

**Fig 3. Spatial tuning of mixed-selectivity cells. (A)** MUA example, with mixed-selectivity, and finely-tuned receptive field. Plots represent firing rates (average across trials ± s.e.) baseline corrected (−500 to −200 ms before cue onset) on individual trial: Exo: blue, Endo: green, Sacc: red. Firing rates at preferred spatial position are significantly different from the other spatial locations for all tasks ($p < 0.005$ or better). **(B)** MUA example, with mixed-selectivity, and board receptive field. All else as in **(A)**. Firing rates at the two preferred spatial positions are significantly different from the other spatial locations for all tasks ($p < 0.05$ or better), except for the Ipsi vs. Contra and Diagonal comparison in the Sacc task. **(C)** Population spatial tuning of mixed-selectivity MUAS with no interaction. Plots represent firing rates (average across MUAs ± s.e.) baseline corrected (−500 to −200 ms before cue onset) on individual trial: Exo: blue, continuous: MUAs with broad receptive fields (RF), dashed: MUAs with finely-tuned receptive fields, Endo: green, Sacc: red. Firing rates at preferred spatial position are significantly different from the other spatial locations for all tasks and conditions ($p < 0.005$ or better). **(D)** Population spatial tuning of mixed-selectivity MUAS with interaction. All else as in **(D)**. Firing rates at preferred spatial position are significantly different from the other spatial locations for all tasks and conditions except for the firing rates of the MUAs with broad receptive fields in the Exo task ($p < 0.001$ or better). Firing rates at Ipsi spatial position are significantly different from the Contra and Diag spatial locations for all tasks and conditions except for the Sacc task ($p < 0.005$ or better). Preferred indicates the spatial location eliciting the highest variation in firing rate from baseline. Ipsi indicates the stimulated spatial position that falls in the same hemifield as the Preferred location. Contra indicates the stimulated spatial position that falls in the contralateral hemifield as the Preferred location. Diag indicates the stimulated spatial position that falls in the contralateral hemifield as the Preferred location, diagonally to it. Data and code to Fig 3 can be found at https://osf.io/z8eh9/.

the impact of the task on the reconfiguration of the MUAs' spatial selectivity, we identified, for all mixed-selectivity MUAs, the two spatial positions eliciting the highest firing rates. We then compared the degree of partial overlap across tasks, pairwise. For the vast majority of mixed-selectivity MUAs without interaction, the receptive fields shared at least one spatial position across each pair of tasks (Exo–Endo: 98%; Exo–Sacc: 95%; Endo–Sacc: 94%). This was also the case for the receptive fields of mixed-selectivity MUAs with interaction, with the marked exception of the Endo-Sacc comparison (Exo–Endo: 96%; Exo–Sacc: 93%; Endo–Sacc: 69%). Because the reconfiguration of receptive fields between the Endo and Sacc tasks was specific to this MUA category, it strongly indicates that these MUAs play a distinct role in these tasks. It is worth noting that task order cannot account for these selective changes, not only because they are specific to this MUA mixed-selectivity category, but also because the order of the Exo and Endo tasks was randomized across different sessions. This possibly relates to previous observations describing shifts of receptive fields in the parietal and prefrontal cortex around saccade execution [46–50]. Likewise, as these shifts are mostly evident when comparing the Endo and Sacc tasks, it would be relevant to understand in what tasks and context such shifts arise [50].

## Task-related information accounts for FEF neuronal variability before cue presentation

In the previous section, we described that the averaged firing rate pattern in response to the cue and during the delay epoch differed as a function of the type of task that monkeys were performing. We hypothesize that such different firing rate patterns as a function of the type of task, result from a context-dependent functional reorganization of the neural population. To test this hypothesis, we used dimensionality reduction to identify the low-dimensional representation of the neural recordings. Dimensionality reduction consists of projecting the high-dimensional neuronal recordings onto a low-dimensional space minimizing information loss. To this end, we applied principal component analysis (PCA) to our data pooled along task category in a pre-cue time interval. At this time of the trial, task-related events soliciting the processing

of spatial information have not been presented yet. Because trials from any given task are organized in blocks, at this time in the trial, the only prior information available to the monkey is task-related information. The projection of this pre-cue neuronal population data onto the principal components of the PCA showed a mixing of task-information across the different components, as task-related neural variability was projected onto multiple PCA axes (S7 Fig). In the following we will use the term task-related information to refer to the information available in the neuronal population about task identity, i.e., about which task the monkey is engaged in.

Other methods of dimensionality reduction allow the labelling of the explained variability by projecting specific sources of variance held in data onto each principal component. One of these methods is the demixed principal component analysis (dPCA) [32]. The objective of this method is to capture the majority of the variance in data in only few latent variables or components (as PCA does) while demixing the dependencies of the population activity on specific parameters by not constraining the components to be orthogonal. This method thus allows to directly interpret the impact of a related feature (e.g., task, position) on the neural population encoding. We applied dPCA on FEF MUA neuronal population activity in order to extract components related to task-related variability, unmixed from other types of variability. To do this, we pooled data across task category (*Exo, Endo,* and *Mem*). This analysis was firstly performed during the pre-cue period (200 ms before cue onset to cue onset) and will be later performed during the cue to target interval (500 ms to 0 ms before target onset) to unmix task- from sensory- specific variance.

The dPCA indicated that 90% of the variance of the recorded brain activity prior to cue presentation was associated with the task (Fig 4A, see S8 Fig for duplication on SUA data). The overall variance explained by dPCA is similar to the variance explained by PCA (Fig 4B) indicating that dPCA reduces dimensionality of the data with a similar performance as PCA, thus minimizing the neuronal information loss. The three first dPCs (i.e., the two first task-related components (#1 and #2), and the first task-independent component (#3)) explained 80% of the variance, indicating that with only few components we recovered the majority of data variance. Therefore, population activity was accurately represented by

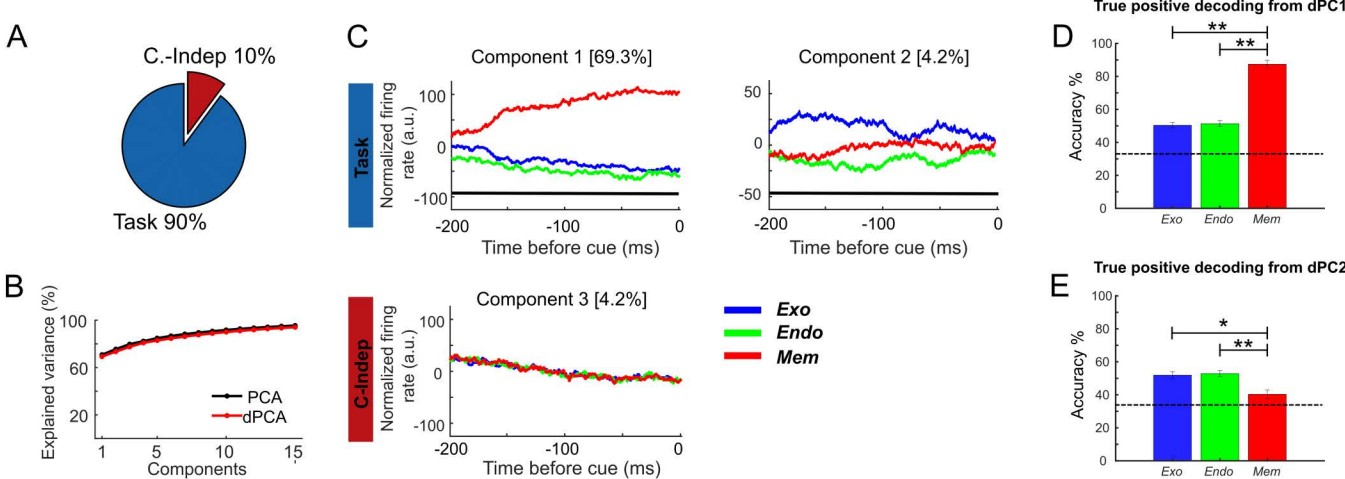

**Fig 4. Demixed PCA unmixes task and independent sources related to variance. (A)** Pie chart shows how the total signal variance is split among parameters: task (blue) and condition independent (red). **(B)** Cumulative variance explained by PCA (black) and dPCA (red). Demixed principal component show similar explained variance as PCA. **(C)** Demixed principal components. In each plot, normalized MUA firing rate averaged per task (in arbitrary units, exogenous (blue); endogenous (green) and memory-guided saccade (red) in the interval from −200 ms to 0 ms locked to the cue onset are projected onto the task-related dPCA components. Thick black lines show time intervals during which the task-information can be reliably extracted from single-trial activity (assessed against 95% C.I.). **(D)** True positive decoding rate for PC1 task-related. **(E)** True positive decoding rate for PC2 task-related. Dashed lines in **(C)** and **(D)** represent the 95% C.I. See S8 Fig for duplication on SUA data. Data and code to Fig 4 can be found at https://osf.io/z8eh9/.

the dPCA components. The angle between the two first components related to the task, did not significantly deviate from orthogonality (88°). This was confirmed by a second analysis performed on individual sessions (mean = 82°, std = 4°). This thus indicated that these two first components represented independent task-related information.

Firing rates projected onto the first dPCA component associated with the task show a dissociation between the two attentional tasks and the working memory task (Fig 4C). Notably, such differences could be due to a reconfiguration of the prefrontal population computations between the different tasks. This reconfiguration could be associated with higher order cognitive factors segregating attentional processes from working memory processes. Alternatively, it could be associated with other task parameters such as effector specificity, the attentional task requiring a manual response, while the memory guided saccade task requires a saccadic response. Further experiments will be needed to precisely identify the computational underpinnings of these task-related differences. The projection of the firing rates onto the second dPCA associated to the task shows a dissociation between the exogenous (blue) and endogenous (green) tasks, suggesting a dissociation as a function of the attentional task-demands specifically within the attentional tasks. S9 Fig, shows, per each session, the projection of the dataset onto each of these components per task (trial-based). The third dPCA is associated with task-independent variance.

In order to assess whether the tuning of each of these two demixed components was statistically significant, we used these components as linear decoders to measure their ability to encode task-related information. To this end, we used cross-validation to measure time-dependent classification accuracy, and shuffling procedure was used to assess the significance of the accuracy measure (see "Methods" for more details about decoding). Confusion matrix of each classification was extracted to analyze the true positive rates (TPR) of the classifier for each task. Fig 4D shows that for the first dPC, the TPR in classifying the memory task was 80% (iqr 7.3), while the TPR for both attentional tasks were significantly lower and not significantly different one from each other (1-way ANOVA, task as main factor: $p = 1.1835e - 15$, Exo versus Mem: post-hoc Bonferroni test: $p = 5.4277e - 14$; Endo versus Mem: post-hoc Bonferroni test: $p = 1.8701e - 14$; Exo versus Endo: post-hoc Bonferroni $p > 0.05$) suggesting a dissociation between attentional and working memory processes. In contrast, TPR in predicting the memory task using the second dPC was lower than predicting both exogenous and endogenous attention (1-way ANOVA, task as main factor: $p = 0.002$, Exo versus Mem: post-hoc Bonferroni: $p = 0.0218$; Endo versus Mem: post-hoc Bonferroni: $p = 0.0023$; Exo versus Endo: post-hoc Bonferroni: $p > 0.05$) suggesting a dissociation in the way the attentional processes (exogenous versus endogenous) were implemented.

In order to confirm that dPC1 reflected a genuine task-identity coding, rather than session order effects, we performed task-dPCA on the independent sessions. For each session and each task, we projected the population activity onto the first task component. We then plotted the average of this activity not as a function of task, but as a function of task order in the session. While the memory guided saccade task was always the last task, half of the sessions started with the Exo task and the other half started with the Endo task. There was no statistically significant difference between the distributions of average activity recorded during the first and second task projected onto the first dPC (S2 File Supporting information). Thus, the task-dPCA component cannot distinguish between first and second task in the session, while it can distinguish between Endo and Exo attentional tasks. These precue effects were not driven by recording instability (S10 Fig and associated text).

## Task-related and spatial information segregate in orthogonal demixed components

Mixed selectivity corresponds to the ability of neurons to simultaneously encode different sources of information [12,13]. The aim of the present analysis was to first investigate whether these different sources of information could be assigned to separate principal components and then to quantify the level of their interaction. Here, we were interested in the task and target spatial information, thus, we focused on the delay period between the cue and the target in order to include spatial information, as opposed to the pre-cue interval that had been selected in our previous analysis, in which there was no spatial information available yet. We applied dPCA on neuronal population activity during the delay epoch, respectively

organized by position independent of task, and by task independent of position, in the time interval corresponding to cue to target interval (500 ms to 0 ms before target onset) (Fig 5, see S11 Fig for replication of analyses on different time intervals and S12 Fig for replication on SUA data). Demixed PCA split variance in position-related variance (52% of the explained variance), task-related variance (26% of the explained variance); and interaction between task information and spatial information (20% of the explained variance) (Fig 5A). The rest of the variance (2%) was related to independent sources of information. The overall variance explained by dPCA was similar to that explained by PCA, both accounting for 70% of the variance by the five first components. As for the pre-cue period (see previous section), we found that the first task-related dPC did not significantly deviate from orthogonality relative to the second task-related dPC during the cue-to-target interval (87.9°), and both of these did not significantly deviate from orthogonality relative to the first position-related dPC (81.6°; and 89.8°). We reproduced these observations when performing this analysis on the independent sessions (dPC1-task versus dPC2-task: mean = 83°, std = 5°; dPC1-pos versus dPC1-task: mean = 74°, std = 8°; dPC1-pos versus dPC2-task: mean = 83°, std = 7°). Therefore, population activity was accurately represented by different orthogonal dPCA components (Fig 5B). Similarly to what we found in the previous section, the two first components associated with the task showed, respectively, a dissociation between attentional task-related processes and working memory task-related process (component 3), and a trend of separation between the two attentional tasks (component 4). In addition, the first component associated with position exhibited a dissociation of each position corresponding to the four expected target locations, whereas the projection onto the second spatial-related component showed two different states, possibly corresponding to the segregation of information across hemifields and recording probes (i.e., probes respectively located into the right or left FEF) (S11 Fig). The distribution of weights attributed to the different units in the first position-related dPC as a function of the first task-related dPC were unimodal, such that most units contribute to the coding of both task- and position-related information (S13 Fig). This supports the fact that mixed-selectivity contributes to the lower dimensional coding of these two variables. Last, the interaction component (Fig 5C, bottom panel) describes the non-linear mixed selectivity between information related to task and position. We used this interaction component to decode trials based on the cued position

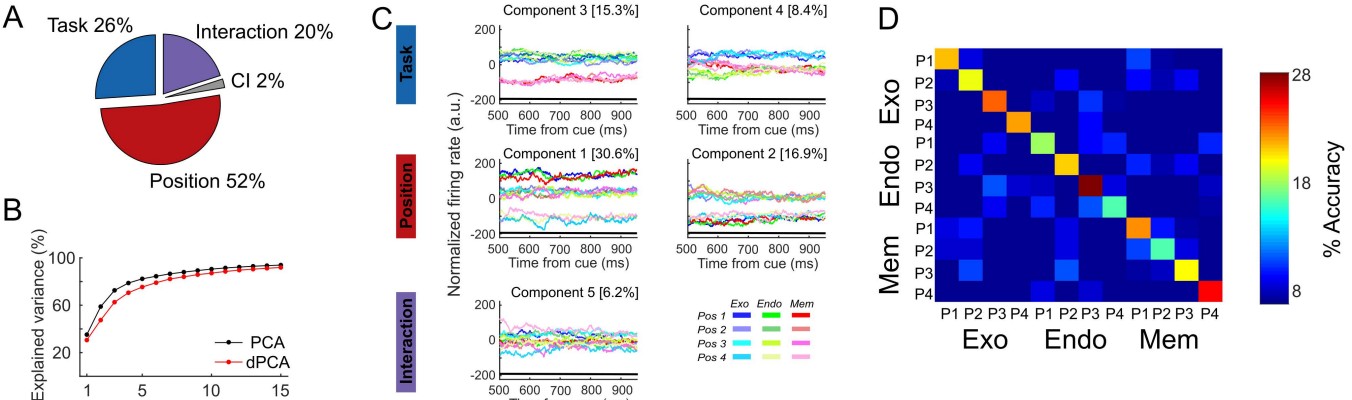

**Fig 5. Demixed PCA unmixes task and spatial sources related to variance. (A)** Pie chart shows how the total signal variance is split among parameters: Task (blue), Position (red), Interaction (purple) and Condition independent (gray). **(B)** Cumulative variance explained by PCA (black) and dPCA (red). Demixed principal components show similar explained variance to PCA. **(C)** Demixed principal components. In each plot, MUA firing rate averaged per task (exogenous (blue shades); endogenous (green shades) and memory guided saccade (red shades)) and cued position in each task, in the interval from 500 ms to 1,000 ms locked to cue onset, are projected onto the task-related, position-related and interaction dPCA components. Thick black lines show time intervals during which the parameter information can be reliably extracted from single-trial activity (assessed against 95% C.I.). **(D)** Confusion matrix of decoding task and position using the demixed interaction components, averaged across all sessions. Absolute chance level of the confusion matrix is 1/12 = 0.08. All elements along the diagonal are significantly above the 95% C.I. Data and code to Fig 5 can be found at https://osf.io/z8eh9/.

and task. Fig 5D shows the average confusion matrix of this decoder across all sessions. We found that true positive rates for each of the 12 classes were significantly above the 95% C.I. (S14C Fig). Overall, this indicates that part of the spatial information represented in the FEF neuronal population is independent (orthogonal) to the task-related information, while another part of this spatial information interacts with the task-related information.

## Discussion

In this manuscript, we addressed the following two questions. First, we asked whether information about the ongoing task identity could be extracted from the FEF. Second, we asked whether the encoding of spatial information in the FEF necessary for the performance of a task depended on task-identity – that is, depended on the cognitive processes underlying the task requirements. To address this question, we analyzed MUA recorded intracranially from the FEF, bilaterally, while two monkeys performed three cognitive tasks in the same recording session. These three tasks presented a very similar structure (the onset of a cue was followed by a target stimulus). However, the attentional tasks required a manual response while the working memory task required a saccadic response and they recruited different underlying cognitive mechanisms necessary for their performance: endogenous spatial attention, exogenous spatial attention, or spatial working memory. The two attentional tasks recruited endogenous spatial attention, exogenous spatial attention processes, in addition to distractor suppression mechanisms and spatial working memory processes. The memory guided saccade task mostly recruited spatial working memory, as well as spatially accurate sensorimotor transformation processes and non-spatialized sensorimotor transformation processes. In the following, we will refer to attention and spatial working memory as the distinguishing features between the two sets of tasks for simplicity. We found that, at the neural level, the activity of FEF cells varied across tasks, suggesting that sensory information is encoded differently depending on the task being performed. Additionally, we found a remarkable population of mixed-selectivity cells that showed a simultaneous encoding of position and task-identity. In order to understand how this information was organized at the population level, we applied a supervised dimensional reduction method (dPCA [31,32]) which unmixed the neuronal variability attributed to different sources of information onto different components. We found that the FEF encoded task identity in two sources of task-related variability (independently from the processing of sensory information). These segregated in orthogonal components that formed a two-dimensional space, with one component dissociating spatial working memory versus spatial attention processes, and the other component dissociating specifically the two spatial attentional processes (endogenous versus exogenous). Additionally, we observed that the neuronal population encoded task-identity and trial-relevant spatial information in a unique interaction component that can be decoded using a single linear readout. This is a hallmark of a high dimensional representation of sensory information. All in all, we show that the FEF has a flexible functional architecture through which it encodes the same spatial information differently as a function of the task demands and underlying cognitive processes.

Prior evidence showed that FEF encoded spatial information during the performance of attentional tasks [5,14,27,37,42,45] and working memory tasks [10,18,20,51,52]. Interestingly, Bahmani and co-workers [52] described a functional overlap between these two cognitive processes in the FEF, and how the coincident impairment of working memory and attention might be associated with disruptions of dopamine signals in this area. Taken together, these previous findings suggest that the FEF is able to encode simultaneously different sources of information, related to sensory or cognitive processes associated with specific tasks. In the present study, we confirm that the FEF neuronal population is able to treat the same sensory information differently as a function of the context set by task identity. This is achieved thanks to a high dimensional representation of the sensory information which endows the FEF with a high degree of flexibility. The neural mechanisms underlying this coding strategy are discussed next.

While our results support the view that the FEF contributes to high-dimensional, task-dependent representations through non-linear mixed selectivity, it is important to consider alternative interpretations of FEF function. Frontal eye field neurons have been shown to be selective to a wide range of signals, ranging from the classical visual and oculomotor

features [14–16], to working memory [17,18], spatial attention [19,20] and attentional or distractibility states [6,42]. These findings have fueled longstanding debates on whether the FEF is primarily a sensory, motor, or cognitive structure. Notably, Lowe and colleagues [53] emphasize the distinction between prefrontal and premotor contributions within the FEF, suggesting that a substantial portion of FEF activity may be more closely tied to the premotor control of saccades rather than abstract cognitive variables. They argue for a more parsimonious account of FEF function, centered on its role in sensory-motor transformations. This perspective raises the possibility that some of the task-related modulation observed in our study could reflect dynamic changes in motor preparation or task-dependent oculomotor contingencies. However, the orthogonal task components identified in our dPCA analyses—particularly during the pre-cue period, when motor planning demands are minimal—point to a more abstract, task-identity-related representation. While we acknowledge the sensory-motor grounding of FEF activity, our data suggest an additional layer of flexible, cognitive encoding consistent with the broader role of the prefrontal cortex in executive control. These two perspectives are not mutually exclusive; rather, their integration may offer the most accurate account of FEF function and the richness of its neuronal computations.

We propose that this flexible encoding ability of the FEF results from the activity of mixed-selectivity cells that are present in the recorded neuronal population, in agreement with previous studies [12,13,42,54]. In our data, we found that 40% of the recorded neurons show simultaneous tuning for position and task-identity. In addition, more than half of the recorded neurons are involved in encoding spatial information in at least two of the tasks performed, and 17% of the recorded cells exhibit spatial tuning in all of the three tasks. Such a rich functional configuration of this area leads to its ability to encode sensory information differently as function of the context (aka, task-identity). As a result, we applied a dimensionality reduction method (demixed PCA, dPCA [31,32]) on the recorded neuronal population in order to split the variance in the data in specific components, which variance is associated to either the sensory information, task-identity and their interaction. Notably, this method is different to the PCA, which decomposes the variance in different components without considering its functional source.

Applying the dPCA during a time-interval prior to cue onset (i.e., corresponding to baseline activity prior to the presentation of the specific ongoing trial spatial requirements, thus reflecting task-related neuronal variability independently from sensory processes, as trials were presented in blocks), we found a two-dimensional subspace of orthogonal components whose activity was linked to task-identity. One of these components dissociated the two main cognitive processes associated with these three tasks (spatial attention variants and spatial working memory), while the second component specifically dissociated the two types of attention in the exogenous task (in which the cue was presented peripherally), and attention in the endogenous task (in which the cue was presented centrally). These results are in line with prior studies [52,55], and show the ability of this area to modulate its patterns of neuronal responses as a function of underlying cognitive process. It is worth noting that in addition, to the fact that attention tasks and memory guided saccade task relied on different cognitive processes, they also differed in that the cued location was indicated by a placeholder in the attentional task, but not in the memory guided saccade task. This can potentially have induced a low-level bias between the two sets of tasks. However, we consider this as unlikely due to the fact that the FEF has been shown to hold a salience map of the most relevant items of the environment. This includes abrupt onset stimuli as well as behaviorally relevant stimuli. Non-informative stable stimuli such as the landmarks, have thus been shown not to be represented in the FEF after their initial processing [56]. As a result, we do not expect these stable landmarks to contribute to the neuronal variance extracted by the dPCA.

A similar pattern of activity was observed after the presentation of the cue, prior to target presentation. That is to say, when dPCA was applied at the interval of time corresponding to the delay period (therefore, when the position information is maintained), we observed the same two-dimensional orthogonal components relative to task identity. This suggests strongly a stable state of the underlying cognitive information encoded by the neuronal population in time, similarly to what has been described in other PFC recordings [42,57,58]. Additionally, a two-dimensional space composed of orthogonal components related to the sensory information was observed specifically during the delay period. This suggests that

FEF simultaneously encodes sensory information and task identity by using different components. Importantly, we found another component related to the interaction between task-identity and the cued position. When we trained a decoder to simultaneously decode task-identity and position information using the firing rates projected onto this interaction component, we found that the accuracy of this classifier was above chance level, indicating that all possible information instances (four cued positions in each task) were decoded. This represents a marker of a high dimensional representation of information in the neuronal population, and it suggests that the FEF specifically encodes the sensory information as a function of the ongoing task – hence as a function of the cognitive process required by the specific task. These results extend those reported by Astrand and colleagues [59], where they showed a different population dynamics and code stability as a function of the source of the information and task. Specifically, they showed, during attentional task dissociating attention orientation from the color and the position of the instruction cues, a stable time-resolved accuracy of decoding spatial attention in attentional tasks, in contrast to the dynamic encoding of color and stimulus position. Other studies from the same team showed a different encoding dynamics of sensory information as a function of whether attention was cued endogenously (centrally) or exogenously (peripherally) [29]. All in all, our results reinforce previous studies suggesting that the dynamics of the encoding of sensory information might be different based on the underlying cognitive process recruited by the task. In addition, we found that the interaction between task and position explained 20% of the variance. This indicates a strong influence of the high-dimensional representation of sensory information on population coding. We decided to focus on the first interaction dPC (accounting for 6.2% of total variance and 20% of variance assigned to the interaction) to check classification accuracies. These were above the absolute chance level (8.33%) as well as above the 95% C.I. Yet they can still be considered as low. An important question is thus whether these interaction components have a functional and/or behavioral impact or not. In a previous study, we show that the degree of interaction impacts overt behavioral performance [42]. How the interaction component reflecting different coding of position across tasks we describe in the present work reflects on behavior is yet unknow. One might for example predict a correlation between the strength of this interaction term and perceptual sensitivity or response criterion (and hence speed) in each task. This will require further investigations in a different set of experiments.

Overall, our findings resonate with recent computational modeling work by Driscoll and colleagues [23], who propose that recurrent neural networks (RNNs), such as those described in prefrontal cortical circuits, utilize shared dynamical motifs—recurring patterns of neural activity implementing specific computations through dynamics such as attractors, decision boundaries, and rotations—to encode task-related information and support flexible multitasking. In their study, RNNs were trained on a diverse set of cognitive tasks commonly studied in humans and animals [24,60,61]. Dynamical motifs clustered in state-space according to task similarity, enabling rapid learning and reconfiguration when faced with task variants. Our empirical evidence of task-state encoding in the prefrontal cortex aligns with these predictions of a modular and compositional neural architecture. Indeed, we describe a lower-dimensional representation of both task identity and task proximity, suggesting a comparable reuse of prefrontal dynamical motifs to enable efficient and flexible task representation and switching, as observed in the RNNs.

Whether such task-dependent sensory encoding that we have observed in the FEF is inherent of its own functional architecture or a consequence of the interaction between its activity and the activity of other brain regions at a network level is something that, unfortunately, we cannot tackle in this study, although is worthy to discuss. Particularly, we have focused on intracranial recordings on the FEF, a key region in the PFC involved in top–down attentional process [62]. However, it is well known that the FEF, given its rich anatomical connectivity, is a functional hub showing efferent and afferent connections to other brain areas such as the dorsolateral prefrontal cortex, the cingulate cortex, the parietal cortex and the superior colliculus, amongst others ([40] for review). In the attention domain, it is known that there is a dual information flow between the FEF and the lateral intra-parietal area (LIP) during the attention-task performance [7]. Other studies have found an active functional role of the LIP during visual working memory tasks [63]. Therefore, one possibility would be that the functional network linking the FEF and the LIP might have a differential information flow regarding the

sensory information, which would impact on the way the FEF encodes top–down sensory signals. Other studies above-mentioned have described the functional role of the prefrontal dopaminergic cells in both attention and working memory [52]. Indeed, dopamine has been shown as a common modulator of attention and working memory, and its imbalance in patients affected by Parkinson's disease have been accompanied by a clear impairment in both cognitive processes. Therefore, another possibility could be that task-identity activity described in the demixed components might be associated with a differential modulation of the activity of these prefrontal dopaminergic cells driven by the task. Another possibility yet would be that these effects of the modulation of the dopaminergic system in the FEF might also impact on the ability to decode the sensory information, reflecting a different encoding strategy of this information based on task-identity. In order to address these hypotheses, further pharmacologic studies based on the dopaminergic prefrontal cortex regulation should be performed.

In conclusion, we show the potential capacities of the FEF in encoding sensory information, and how this information encoding is performed differently as a function of the cognitive demands of the underlying task. We believe that these results provide novel insights on how the FEF organizes, from a computational perspective, attention and working memory information and recruits them in the context of specific tasks. This knowledge may also guide pharmacological interventions can be targeted to improve these cognitive processes in patients affected by neurological diseases. Moreover, this knowledge leads to an improvement in the design of cognitive brain-computer interface (cBCI) field, opening the venue to the manipulation of the activity not only related to sensory or motor processes, but also to specific cognitive modalities [64–66].

## Supporting information

**S1 File. Stability of MUA/SUA selectivity across trials.** Task-related selectivity might be accounted for by changes in the signal along the recording session, resulting in possibly different firing rate fluctuations not related with the cognitive aspects of the tasks. To reduce this effect, we decided to work with normalized activity with respect to the baseline in each trial. Furthermore, to prove that our selectivity results are not driven by systematic firing rate fluctuations driven by the time in the session, we conducted a supplementary analysis in which we extracted the task and position selectivity by selecting trials randomly along the recording session. In this way, our data did not contain any temporal structure given by the time-course along the recording session, nor information about task or position. We performed this procedure 100 times and computed the % of position, task, mixed selectivity and non-selective channels. The 95% confidence interval for both task and position are below 2.5% for all categories. We conducted a second analysis in which we computed the % of position, task, mixed selectivity and non-selective channels over a randomly selected subset of 10 trials for each possible trial category. We repeated this procedure 100 times. The outcome of this analysis is very close to that reported in main Fig 2. These supplementary analyses are captured by the figure below. Over all, these additional analyses indicate that the reported results in main Fig 2 are robust to neuronal variability as well as to possible temporal structure in the task. Barplot corresponding to the proportion of MUAs (left) or SUAs (right) tuned to only position (red), task (blue) and both (additive mixed-selectivity; yellow; mixed selectivity with interaction; green, mean± s.e. computed over 100 repetitions of over a random selection of 10 trials for each possible trial category). Non-selective MUAs/SUAs are plotted in gray. Black dashed line corresponds to the 95% CI computed independently for each category of cells. Note that the average proportion of non-selective MUAs/SUAs is very low relative to the 95% CI estimated from a random permutation procedure. Data and code to figure SM1 can be found at https://osf.io/z8eh9/. (PDF)

**S2 File. Despite the precautions taken to stabilize the electrodes after their insertion in the brain, and trial-based baseline correction, a drift related to physiological parameters, could still impact neuronal recordings along the session.** In contrast with other methods such as decoding or PCA, the analysis of neuronal variability that we implemented in this work relies on demixed principal component analysis (dPCA, see below). This method has been shown in

the literature not to be affected by the drift in the signal (Kobak and colleagues [32]). Indeed, it extracts the neuronal variance associated to specific parameters (here, task or position), and separates this neuronal variance from the parameter-independent variance. Monkeys performed each task for 20 min or more. Thus, drift could happen both within any task and between tasks. These changes in neuronal responses were captured in the parameter-independent variance. We further validate that drift did not account for the main results presented in Figs 4 and 5, as follows. We performed task-dPCA on each of the independent sessions. For each session and each task, we projected the population activity on the first task-component. We then plotted the average of this activity not as a function of task, but as a function of task order in the recording session. While the memory guided saccade task was always the last task in the session, half of the sessions started with the Exo task and the other half started with the Endo task. We found that there was no statistically significant difference between the distributions of average projected activity recorded during the first and second task onto the first dPC (figure below). Thus, the task-dPCA component cannot distinguish between first and second task in the session, while is can distinguish between Endo and Exo attentional tasks. Normalized firing rate of projected population activity on the first task-component of the dPCA per session, computed independently for the first task in the session (50% Exo, 50% Endo), the second task in the session (50% Exo, 50% Endo) and the last task in the session (100% memory-guided saccade task; 1-way non-parametric ANOVA: $p = 5.0396e - 14$; post-hoc Bonferroni test, task 1 versus task 2: $p = 1$, task 1 versus task 3: $p = 6.2112e - 13$, task 2 versus task 3: $p = 2.2593e - 12$). Please note that this analysis cannot be performed on SUA data as it requires dPCA to be performed at the session level. Data and code to figure SM2 can be found at https://osf.io/z8eh9/.
(PDF)

**S1 Fig. Exemplar waveforms for spike sorted single units (SUA) sampled on different channels and sessions.** Mean (±s.e.) of each waveform is presented per task (Blue: Exogenous attentional task; Green: Endogenous attentional task; Red: Memory guided saccade task). Data and code to S1 Fig can be found at https://osf.io/z8eh9/.
(TIFF)

**S2 Fig. Mixed selectivity corresponds to the ability of a neuronal population to encode simultaneously independent sources of information: (A) Firing rate of a representative SUA aligned on the cue (200 ms before the cue onset to 1,000 ms after the cue onset) in exogenous task (blue), endogenous task (green) and memory guided saccade task (red) for preferred (solid line) and non-preferred (dash line) position.** The data were baseline corrected (−500 to −200 ms before cue onset) on individual trial and channel. Gray box corresponds to the delay period during which neuronal activities were analyzed for subplots C and D. **(B)** Firing rate of a representative SUA (same as in **A**) aligned on the target (500 ms before the target onset to 500 ms after the target onset). All else as in **A**. **(C)** Mean firing rate of the SUA neuronal population in the three tasks aligned on cue presentation (200 ms before the cue onset to 1,000 ms after the cue onset) in the exogenous task (blue), the endogenous task (green) and the memory guided saccade task (red) for preferred (full line) and non-preferred (dash line) position. The data were baseline corrected (−500 to −200 ms before cue onset) on individual trial and channel. Gray box corresponds to the delay period during which neuronal activities were analyzed for subplots C and D. **(D)** Firing rate of the SUA neuronal population (same as in **C**) aligned on the target (500 ms before the target onset to 500 ms after the target onset). All else as in **C**. **(E)** Venn diagram representing the proportion of cells showing a spatial tuning in each of the three tasks. **(F)** Pie chart corresponding to the proportion of neurons tuned to only position (red), task (blue) and both (Additive mixed-selectivity; yellow; Mixed-selectivity with interaction; green). Non-selective neurons are plotted in gray. Wilcoxon ranksum test was performed between baseline (−100 to 0 before the cue onset) and time interval (800–900 ms after the cue onset) for each channel and position across trials. A non-parametric 2-Way-ANOVA was performed on selective channels with Bonferroni–Holm correction on spatial and task factor on time interval from 600 to 900 ms after the cue onset ($p < 0.01$). Data and code to S2 Fig can be found at https://osf.io/z8eh9/.
(TIFF)

**S3 Fig. (A) Firing rate of a representative MUA (same as in Fig 2) aligned to target onset (400 ms before target onset to 600 ms after target onset) averaged across trials of the exogenous attention task (blue), the endogenous attention task (green) and the memory guided saccade task (red) for preferred (solid line) and least-preferred (dash line) positions.** The data were baseline corrected (−500 to −200 ms before cue onset) on individual trials and channels. **(B)** Mean firing rate across the neuronal population (MUA) aligned on the target (−400 ms before target onset to 600 ms after target onset) as a function of tasks and positions (same as in **A**). The data were baseline corrected (−500 to −200 ms before cue onset) on individual trials and channels. Data and code to S3 Fig can be found at https://osf.io/z8eh9/.
(TIFF)

**S4 Fig. (A) MUA example of task selectivity (left), position selectivity (middle) and mixed-selectivity (right). (B)** Population activity for task selective MUA (left), position selective MUA (middle) and mixed-selectivity MUA (right). Gray box corresponds to the delay period during which neuronal activities were analyzed for subplots C and D of Fig 2. **(C)** SUA example of task selectivity (left), position selectivity (middle) and mixed-selectivity (right). **(D)** Population activity for task selective MUA (left), position selective MUA (middle) and mixed-selectivity MUA (right). Gray box corresponds to the delay period during which neuronal activities were analyzed for subplots E and F of S2 Fig. Data and code to S4 Fig can be found at https://osf.io/z8eh9/.
(TIFF)

**S5 Fig. (A) MUA: Top row: Venn diagram representing the proportion of cells showing a spatial tuning in each of the three tasks.** Bottom row: pie chart corresponding to the proportion of neurons tuned to only position (red), task (blue) or both (mixed-selectivity; yellow) during different periods in the trial (Precue: from 0 to 500 ms before the cue onset; Early Cue: from 0 to 500 ms after the cue onset; Delay: from 500 to 1,000 ms after the cue onset; Pre Target: from −500 to 0 ms before the target onset; and Target: from 0 to 500 ms after the target onset). Proportion of non-selective neurons is plotted in gray. This figure complements Fig 2C and 2D. **(B)** SUA: all as in **A**. This figure complements S2E and S2F Fig. Data and code to S5 Fig can be found at https://osf.io/z8eh9/.
(TIFF)

**S6 Fig. Spatial tuning of mixed-selectivity cells. (A)** MUA example, with mixed-selectivity without interaction, and finely-tuned receptive field (Exo: blue, Endo: green, Sacc: red). **(B)** MUA example, with mixed-selectivity without interaction, and board receptive field. **(C)** MUA example, with mixed-selectivity with interaction, and finely-tuned receptive field (Exo: blue, Endo: green, Sacc: red). **(D)** MUA example, with mixed-selectivity with interaction, and board receptive field. **(E)–(H)** Same as **(A)–(D)** for SUA example. Data and code to S6 Fig can be found at https://osf.io/z8eh9/.
(TIFF)

**S7 Fig. Projection of MUA population activity onto the first and second principal components extracted with a PCA analysis.** Each point corresponds to the projection of the activity in each trial (averaged in the time interval −200 to 0 ms pre Cue, each circle thus corresponds to one trial) in each task (Exogenous task, blue; Endogenous task; green; Memory saccade task, red). Each plot corresponds to a different session. Plots 1–7 are from monkey 1 and plots 8–12 are from monkey 2. Data and code to S7 Fig can be found at https://osf.io/z8eh9/.
(TIFF)

**S8 Fig. Demixed PCA unmixes task and independent sources related to variance in SUAs. (A)** Pie chart shows how the total signal variance is split among parameters: task (blue) and condition independent (red). **(B)** Cumulative variance explained by PCA (black) and dPCA (red). Demixed principal component show similar explained variance as PCA. **(C)** Demixed principal components. In each plot, normalized MUA firing rate averaged per task (in arbitrary units, exogenous (blue); endogenous (green) and memory-guided saccade (red) in the interval from −200 ms to 0 ms locked to the

cue onset are projected onto the task-related dPCA components. Thick black lines show time intervals during which the task-information can be reliably extracted from single-trial activity (assessed against 95% C.I.). **(D)** True positive decoding rate for PC1 task-related. **(E)** True positive decoding rate for PC2 task-related. Dashed lines in **(C)** and **(D)** represent the 95% C.I. Note that the decoding presented in Fig 4 could not be reproduced here, due to the fact that SUA data was not extracted for all recording sessions. Data and code to S8 Fig can be found at https://osf.io/z8eh9/.
(TIFF)

**S9 Fig. Projection of MUA onto the first and second demixed principal components extracted with a dPCA analysis.** Each point corresponds to the projection of the activity in each trial (averaged in the time interval −200 to 0 ms pre-Cue, each circle thus corresponds to one trial) in each task (Exogenous task, blue; Endogenous task; green; Memory saccade task, red). Each plot corresponds to a different session. Plots 1–7 are from monkey 1 and plots 8–12 are from monkey 2. Data and code to S9 Fig can be found at https://osf.io/z8eh9/.
(TIFF)

**S10 Fig. Raster plots of representative MUA and SUA examples. (A)** Raster plot of a representative MUA channel showing spike activity throughout an entire session (First task in session, Endo: green; Second task, Exo: blue; Third task, Sacc: red). Firing rates are shown below the raster plots, aligned to cue onset (−200 ms to +1,000 ms) and to target onset (−500 ms to +500 ms). **(B)** Raster plot of a representative SUA example, as in **(A)**. First task in session, Exo: blue; Second task, Endo: green; Third task, Sacc: red. The inset shows the waveform of the selected spike-sorted unit from the same channel. Data and code to S10 Fig can be found at https://osf.io/z8eh9/. Focusing on spiking activity during the precue period (−200 to 0 ms relative to cue onset), we observed significant variation in firing rates across the three tasks, consistent with the task-related components identified by the dPCA during this epoch (Fig 4). To assess whether these differences might be explained by variations in trial-to-trial spiking variability—potentially due to recording instability—we computed the coefficient of variation (CV) of spike counts across trials, separately for each MUA and SUA, for each task and session. For MUAs, median CVs were: $Exo = 0.2460 \pm 0.0185$, $Endo = 0.2224 \pm 0.0172$, and $Sacc = 0.2535 \pm 0.0201$. For SUAs, median CVs were: $Exo = 0.8687 \pm 0.0572$, $Endo = 1.1808 \pm 0.0957$, and $Sacc = 0.9688 \pm 0.0625$. Non-parametric pairwise comparisons revealed no significant differences in CV across tasks for either MUAs or SUAs (all $p > 0.05$). These findings indicate that the observed task-dependent changes in spiking rates cannot be attributed to differences in inter-trial variability. Instead, they support the conclusion that task identity modulates neural activity during the precue period in a structured and reliable manner.
(TIFF)

**S11 Fig. All as in Fig 4, except that dPCA is performed on a time interval running from (A1 to D1) 0–500 ms after cue presentation; (A2 to D2) −500 to 0 Pre-target presentation and (A3 to D3) 0–500 ms Post-target presentation.** Absolute chance level of the confusion matrix is $1/12 = 0.08$. Data and code to S11 Fig can be found at https://osf.io/z8eh9/.
(TIFF)

**S12 Fig. Demixed PCA unmixes task and spatial sources related to variance in SUAs. (A)** Pie chart shows how the total signal variance is split among parameters: Task (blue), Position (red), Interaction (purple) and Condition independent (gray). **(B)** Cumulative variance explained by PCA (black) and dPCA (red). Demixed principal components show similar explained variance as PCA. **(C)** Demixed principal components. In each plot, MUA firing rate averaged per task (exogenous (blue shades); endogenous (green shades) and memory guided saccade (red shades)) and cued position in each task, in the interval from 500 ms to 1,000 ms locked to cue onset are projected onto the task-related, position-related and interaction dPCA components. Thick black lines show time intervals during which the parameter information can be reliably extracted from single-trial activity (assessed against 95% C.I.). **(D)** Confusion matrix of decoding task and position

using the demixed interaction components, averaged across all sessions. Absolute chance level of the confusion matrix is 1/12 = 0.08. All elements along the diagonal are significantly above the 95% C.I. Data and code to S12 Fig can be found at https://osf.io/z8eh9/.
(TIFF)

**S13 Fig. Distribution of weights attributed to the different units in the first position-related dPC as a function of the first task-related dPC, for both SUAs (left) and MUAs (right).** Data and code to S13 Fig can be found at https://osf.io/z8eh9/.
(TIFF)

**S14 Fig. (A) Barplot corresponding to the true positive rates obtained using the first task-related demixed principal components as a linear classifier to classify trials by task (Endogenous attention, blue; Exogenous attention, green; Memory saccade, red) (1-way ANOVA, task as main factor, on dPC1: $p = 9.6184e − 10$: Exo versus Endo: post-hoc Bonferroni: $p > 0.05$; Exo versus Mem: post-hoc Bonferroni: $p = 1.8546e − 8$; Endo versus Mem: post-hoc Bonferroni: $p = 8.6775e − 9$; 1-way ANOVA, task as main factor, on dPC2: $p = 4.0943e − 6$, Exo versus Endo: post-hoc Bonferroni: $p > 0.05$; Exo versus Mem: post-hoc Bonferroni: $p = 0.0011$; Endo versus Mem: post-hoc Bonferroni: $p = 3.1361e − 6$). (B)** Barplot corresponding to the true positive rates obtained using the first position-related demixed principal components as a linear classifier to classify trials by position (1-way ANOVA, position as main factor, on dPC1: $p = 0.0367$; post-hoc Bonferroni: all positions comparisons showed no significant difference, $p > 0.05$; 1-way ANOVA, position as main factor, on dPC2: $p = 0.4449$). **(C)** Barplot corresponding to the true positive rates obtained using the first task × position interaction demixed principal component as a linear classifier to classify trials by position in each task (2-way ANOVA, position × task; task: $p = 0.7971$; position: $p = 0.8539$; interaction: $p = 0.4104$). In all plots, black dashed lines indicate absolute chance level. Gray spaced dashed lines indicate 95% C.I. Data and code to S14 Fig can be found at https://osf.io/z8eh9/.
(TIFF)

## Acknowledgments

We thank engineer Serge Pinède for technical support, Jean-Luc Charieau and Fabrice Hérant for animal care, and Johan Pacquit, Sylvain Maurin and Thomas Perret for informatics assistance. All procedures were approved by the local animal care committee (C2EA42-13-02-0401-01) in compliance with the European Community Council, Directive 2012/63/UE on Animal Care.

## Author contributions

**Conceptualization:** Claire Wardak, Julian Luis Amengual, Suliann Ben Hamed.

**Data curation:** Axel Mouille, Elaine Astrand.

**Formal analysis:** Axel Mouille, Elaine Astrand, Julian Luis Amengual.

**Funding acquisition:** Suliann Ben Hamed.

**Investigation:** Elaine Astrand, Claire Wardak, Suliann Ben Hamed.

**Methodology:** Julian Luis Amengual, Suliann Ben Hamed.

**Project administration:** Suliann Ben Hamed.

**Resources:** Axel Mouille, Corentin Gaillard, Elaine Astrand, Julian Luis Amengual, Suliann Ben Hamed.

**Supervision:** Julian Luis Amengual, Suliann Ben Hamed.

**Validation:** Julian Luis Amengual, Suliann Ben Hamed.

**Visualization:** Axel Mouille, Julian Luis Amengual.

**Writing – original draft:** Axel Mouille, Julian Luis Amengual, Suliann Ben Hamed.

**Writing – review & editing:** Axel Mouille, Corentin Gaillard, Elaine Astrand, Claire Wardak, Julian Luis Amengual, Suliann Ben Hamed.

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
