## [Editor Report · Decision Letter 0]

15 Nov 2023

Dear Dr Ben Hamed,

Thank you for submitting your manuscript entitled "Distinct neural states encode task identity in frontal eye field and interact with its core spatial properties" for consideration as a Research Article by PLOS Biology.

Your manuscript has now been evaluated by the PLOS Biology editorial staff as well as by an academic editor with relevant expertise and I am writing to let you know that we would like to send your submission out for external peer review.

Once your full submission is complete, your paper will undergo a series of checks in preparation for peer review. After your manuscript has passed the checks it will be sent out for review. To provide the metadata for your submission, please Login to Editorial Manager (https://www.editorialmanager.com/pbiology) within two working days, i.e. by Nov 17 2023 11:59PM.

Kind regards,

Christian

Christian Schnell, PhD

Senior Editor

PLOS Biology

cschnell@plos.org

---

## [Decision Letter · Decision Letter 1]

17 Jan 2024

Dear Dr Ben Hamed,

Thank you for your patience while your manuscript "Distinct neural states encode task identity in frontal eye field and interact with its core spatial properties" was peer-reviewed at PLOS Biology. It has now been evaluated by the PLOS Biology editors, an Academic Editor with relevant expertise, and by two independent reviewers.

In light of the reviews, which you will find at the end of this email, we would like to invite you to revise the work to thoroughly address the reviewers' reports.

As you will see below, the reviewers are quite supportive but request some clarifications, more methodological details, and further analyses.

Given the extent of revision needed, we cannot make a decision about publication until we have seen the revised manuscript and your response to the reviewers' comments. Your revised manuscript is likely to be sent for further evaluation by all or a subset of the reviewers.

**IMPORTANT - SUBMITTING YOUR REVISION**

*Re-submission Checklist*

*Published Peer Review*

*PLOS Data Policy*

*Blot and Gel Data Policy*

Sincerely,

Christian

Christian Schnell, PhD

Senior Editor

PLOS Biology

cschnell@plos.org

REVIEWS:

Reviewer #1: In this manuscript, Mouille et al examine how neuronal activity in the Frontal Eye Field (FEF), in the prefrontal cortex of monkeys, encodes task identity and spatial information. To this end, they have employed multiunit activity recorded from the two hemispheres, during three different tasks: two cued spatial attention tasks and a memory guided saccade task. There are three main findings: a) task identity is encoded by FEF activity at the population level during fixation and before the onset of the spatial cue, b) several FEF sites encode both task identity and spatial information following the cue, some also in a non-linear manner and c) different components of population activity in the FEF can discriminate on one hand task context and on the other hand spatial information in different tasks. They suggest that this pattern of mixed selectivity results in high dimensional representations that can encode flexibly and accurately behaviorally relevant information according to current demands.

The analyses are sound and interesting and support the authors' conclusions. Conceptually, the paper provides convincing evidence and innovative findings to answer a timely question: how different types of behaviorally relevant information including task, are represented in the brain within a single cortical area. Thus, I believe the paper advances our current knowledge. I do, however, have some concerns and suggestions that can hopefully help clarify some of the methods, results and their presentation. I list these below:

MAJOR

1) I am confused about the summary of results provided in the concluding paragraph of the Introduction. It is written that "spatial information is represented independently from task-related information, indicating that task-identity is represented at the same hierarchical level as specific task parameters such as position". Presumably this describes the fact that in the demixed PCA analysis different components were largely found to explain the variance for task vs spatial information. However, the authors did find a component of activity that encoded the interaction of the two variables, which would indicate that task identity and spatial information are not encoded independently in absolute terms. Please, explain.

2) There are a number of points that are not clear in the Methods.

a) In the description of the attention tasks, the authors refer to a change in luminosity of targets or distractors that the monkeys should respond to. Was that an increase or decrease in luminosity and what were the exact levels of luminance before and after the change. Although this was similar in the two attention tasks, it is critical to appreciate the difficulty of detecting the change and thus the required cognitive load in each task.

b) The data preprocessing for multi unit activity is somewhat confusing. It is mentioned that "epoched MUA data were baseline corrected (independently for each channel and each trial) relative to the period from -500 ms to -300 ms before the cue". Does this refer to the firing rate plots shown in Figure 2A,B? This doesn't seem to be the case, yet it is unclear what kind of normalization was done in these plots. I am not sure why correction was applied in each trial rather than the average across trials. If on the other hand, this refers to preprocessing for the PCA or dPCA analysis this should be clearly stated and a description of normalization for data shown in plots Fig 2A,b should be added.

c) To assess spatial selectivity the authors used a Wilcoxon test comparing activity in the pre-cue and post-cue period. Was this a signed rank test? What was the p-value used to classify a channel as selective? Was activity for each channel averaged across all four cued positions? I do not understand how this comparison examines spatial selectivity. Comparison with baseline does not compare across positions and may have over- or underestimated spatial selectivity by including visually responsive channels rather than spatially selective channels. An ANOVA or a Kruskal Wallis across all four positions in the post cue period would be more appropriate.

3) In the Results, in the first paragraph of page 8, it is stated that "Although these three tasks showed some similarities in their structure, they recruited different cognitive processes". However, differences are not limited to cognitive demands. The sensory conditions are also very different and this should be clearly stated. Particularly, when comparing the memory guided saccade task to the attention tasks, the absence of stimuli in the former imposes a critical difference that could in principle explain the results beyond cognitive demands. It is the fact that differences across the two attention task that are more similar in structure are found that allows the authors to generalize their conclusions about task identity encoding. I believe this should be discussed.

4) In the last paragraph of page 8 it is stated that both monkeys showed lower false alarm rates in the endogenous task compared to the exogenous task, but the opposite is shown in Figure 1B.

5) On page 10, the authors report the proportion of channels that were found to be selective for position only, task only and both task and position. I am not sure how to interpret these numbers. Given the variability in firing rates, even across trials, I can think of potential confounding factors that would need to be ruled out. For example, is it possible that task-related selectivity is a result of changes in the signal over time given that the tasks were run in blocks? How do these numbers e.g. for position selectivity or mixed selectivity compare to those expected by chance (by shuffling trials)?

6) On page 12, second paragraph, a 2-way ANOVA is mentioned but the two factors are not mentioned and neither are the p-values for the two factors. Only one p-value is reported. The same is true in several other places in the text (e.g. three lines below, legend of supplementary figure 4)

7) The authors used the interaction component (position x task) to decode position and task information. Although accuracies are significantly above chance, they are very low. What are the implications of this finding for the authors' conclusions? How do such low accuracies fit into the context of high dimensional representations subserved by mixed selectivity? A more detailed discussion is needed.

8) In the discussion (first paragraph) the authors write: "the dynamics of the activity of the FEF cells when their preferred position was cued was different across tasks…". Why not also when their least preferred position was cued? None of the analyses was specific for the cued position, was it? This is rather confusing.

9) In the legend of Supplementary Figure 2 when referring to "Projection of MUA" it should be clarified that this is population activity

10) In the legend of Supplementary Figure 4, in B and C there are no statistics for the comparison to chance

MINOR

1) In Figure 1 where the tasks are shown, the last part of the memory guided saccade task, where according to the description in the methods, the monkey waited for the peripheral stimulus to change color from red to grey is not shown

2) Page 8 first line: replace "avoid" by "ignore"

3) Page 8, penultimate paragraph, last sentence: "in despite" delete "in"

4) There are a few instances where the authors do not consistently use either past or present tense in the same sentence. For example, on page 10, first line: "…these recorded neurons SHOW enhanced responses when the cue WAS presented…". Similarly on page 15, 5 lines before the end: "These results ARE in line with prior studies (Bahmani et al., 2019; Panichello and Buschman, 2021), and SHOWED…"

5) In the same sentence rephrase "…and this for all three tasks" by e.g. and this was true for all three task

6) The title in Figure 2 should change to be more specific and reflect the content

7) Page 10, line 4: "24% of cells were tuned to the three different tasks". Do the authors mean in all three tasks

8) In the same sentence: "suggesting that there is a neural population encoding multiple cognitive processes (spatial attention & spatial working memory)". Please rephrase.

9) On page 11, first paragraph: "Dimensionality reduction consists of projecting…" use either consist in or rephrase

10) Page 12, line 5: replace show by shows

11) Page 12, four lines before the end: "using the dPC" do the authors mean using the second dPC?

12) Page 13, line 6: "we will focused" use either we focused or we will focus

13) Legend of Figure 4: "per tasks" replace by per task

14) Page 14, four lines before the end: "dissociating specifically both spatial attentional processes" replace both by the two.

15) Page 16, first paragraph: "These results extended…" replace extended by extend

Reviewer #2: This is a neurophysiological study in macaque monkeys that compared 3 different tasks (memory-guided saccade task and two spatial attention / detection tasks with manual responses) in the same recording sessions, in order to answer the question whether there is task identity representation besides the representation of task-relevant parameters such as spatial position selectivity, in the frontal eye fields (FEF). I have read it with great interest. The results show that indeed, the task identity is represented in FEF and the two attentional tasks are represented similarly to each other but differently from the memory-guided saccade task. Furthermore, there is an interaction between spatial selectivity and the task-identity selectivity after the spatial information is available, suggesting a high-dimensional representation of spatial factors that might facilitate flexible adjustment to specific task demands.

These results are of substantial interest and, as far as I can judge, are novel for the FEF and might constitute a significant advance to warrant a publication in PLoS Biology. I have however a number of concerns or questions related to insufficient methodological descriptions, experimental design, and presentation of the results, as well as suggestions for additional analyses, which should be addressed in a revision.

Page 6

The electrophysiological methods are not sufficiently clear and need to be explained better and more comprehensively:

"A threshold defining the MUA was applied independently for each recording contact and before the actual task-related recordings started." - this is not clear, please describe the procedure in more detail and explain why the threshold has been defined online and not during the data analysis stage offline, taking into account recording SNR etc. Please explain how the MUA signal was derived (e.g. as envelope of the MUA energy, or the actual threshold crossings resulting in "multiunit spike trains" - as it seems to be the case) and what were the thresholds relative to some measure of the signal SNR - e.g. in terms of number of standard deviations of the activity.

Furthermore, the exclusive emphasis on multiunit activity is surprising, especially having in mind the putative mixed selectivity properties. It seems that for the aims of the current study, carefully spike-sorted single neurons would be better, at least when the signal quality would allow reliable spike sorting. It seems that there was no attempt to pursue proper spike sorting. Please explain/justify the choice of this approach. I would highly recommend to supplement the results of MUA with single neuron analysis.

Please present number of sessions and the overall number of recording sites/channels for each animal and task, as well as the number of trials for each task in each session.

Please describe how the stability of the recording was assessed, and what measures were taken to ensure that the differences observed between task blocks are not attributable to recording instability or drift, which is often observed with acute U-probe recordings.

"Spatially selective channels were identified using a non-parametric Wilcoxon test of the activity between

the baseline and a post cue period (1300 to 1400 ms following cue onset)." - Perhaps I am missing something obvious, but I don't understand what is especially relevant for the spatial selectivity in this short 100 ms interval long after the cue onset. Is this a proxy for the delay period activity? But this would not work for shorter delay periods…

"A two-way ANOVA Task x Position" - was this a non-parametric two-way ANOVA (given the non-parametric Wilcoxon test of the activity between the baseline and a post cue period)? Please specify/justify.

Please explain how was MUA normalized (e.g. for the Figure 2A,B and further analyses). It is mentioned that "To extract the averaged firing rates, these epoched MUA data were baselinecorrected (independently for each channel and each trial) relative to the time period from -500 ms to -300 ms before the cue.", but baseline subtraction would not result in the arbitrary units shown in Figure 2. What are these units, and why the activity is not shown in Hz, as in Figures 3 and 4?

In terms of the task design, I am surprised the authors did not match the spatial display of the memory-guided task to the other two tasks better, by placing the 4 placeholders identical to the attention tasks target/distractors and indicating the actual saccade target with a brief cue. Without this matching, the tasks differ not only in cognitive demands, but also in the low-level visual stimulation. This might represent a potential confound, since the task-selectivity might at least partially stem from the sensory spatial information, even before cue onset. Please explain this design choice and provide counter-arguments as to why this does not invalidate the results on the task encoding. It seems that there were only 4 possible positions in each session anyway, so showing 4 placeholders would not invalidate the spatial memory component of the actually cued location.

Results

Page 8

It seems counterintuitive that the hit rate in M1 was lowest for arguably the simplest Mem task. Why was this task so demanding for M1? This also seems to be in conflict with the statement in page 6 "while the memory-guided saccade task was always presented in the third order, as monkeys displayed a strong behavioural preference for this task." - how was this preference assessed?

Please explain the usage of the nonparametric Friedman test here (i.e. what are repeated measures, what are degrees of freedom, is this analysis across sessions, target locations, etc).

Page 10

"Indeed, we found a significant task-effect on the response to the preferred position (nonparametric

Friedman test χ2(5) = 39.3, p = 2.89e-09)." Please explain the usage of the nonparametric Friedman test here (what are repeated measures, what are degrees of freedom, etc). Is this statement about the example channel in Figure 2A, or the entire population? How is this related to the two-way ANOVA on the task and position?

Please illustrate the channels of all 3 types. I assume the channel illustrated in the Figure 2A is "mixed selective", please show also only spatial position-selective and task-selective channels. I would also strongly recommend splitting the population into the 3 categories and showing both, the example channel and the population, for all 3 cases.

Figure 2C,D - from which trial periods these analyses are derived? I was expecting the same period as the dPCA analysis (500 ms to 0 ms before target onset) for consistency (but the later comments on other task intervals).

Did channels showing exclusively task selectivity in the interval after the spatial information was available - i.e. at some point after the cue - also show task selectivity in the pre-cue interval? Please relate task identity selectivity across these two intervals.

Generally, I think the results in the Figures 3 and 4 are overall interesting and compelling, but I find a considerable disconnect between the basic - usual standard statistical methods - but largely incomplete analysis in the Figure 2 and the rest, in terms of the time intervals and the level of description. Furthermore, I see no convincing reason to focus only on a single, late delay interval, especially because the analysis in figures 3 and 4 is largely time-resolved anyway. I suggest to expand this analysis to at least two more epochs - pre-cue (as is already done), right after the cue, late delay (as is already done), and response period (e.g. 0 - 500 ms after target onset, or aligned to saccade/manual response) - to capture selectivity profiles of most task-relevant intervals. In fact, I hope that the analysis of the post-cue/early delay period could more consistently highlight the difference / task identity selectivity between the two attentional tasks, because the effect of the exogenous vs endogenous cueing could be more pronounced in the early delay period compared to the late delay where the origin of the attentional cue is already of little relevance.

Regarding the interpretation of task identity selectivity, I find the argument "To note, such difference cannot be interpreted as related to the response movement effector (hand response or saccade), since data was analysed during pre-cue activity (prior to motor preparation)." flawed. Because of the block design, monkeys could (and likely did) mobilize the appropriate effectors in a sustained fashion, and perhaps adjusted posture. I think that a large proportion of task selectivity, concerning the most pronounced difference between Mem and manual response attentional tasks, might stem from effector specificity, and I don't think authors can easily dismiss this interpretation. Furthermore, the above argument clearly cannot be applied to the delay period interval, but I don't think the authors try to claim completely different mechanisms of task selectivity for the pre-cue and the delay intervals. Importantly, I don't think this possibility invalidates/compromised the findings in any way - whether the difference between the tasks is purely in the "cognitive" or/and in the "effector"-specific domain, it is still very interesting.

Partially related to this, I think that spatial working memory process plays a role in all 3 tasks, and not only in the Mem task. Note that the monkeys had to remember which of the 4 locations was cued, in order to ignore distractors and respond to the target. I fully agree that the two attentional tasks present additional cognitive demands compared to the Mem task, but I think the spatial working memory is a largely shared process here, and should not be highlighted as a major discriminant.

Minor:

Page 3: I found the first paragraph of the Introduction quite vague and long-winded, and it did not help me to set up the research question. It also was not convincing stylistically (e.g. 2nd sentence "To do

so, …"; 3rd sentence "In doing so,…" and hard to follow. Please consider rewriting.

"in the representation of specific task-parameters such as position, color or attention irrespective

of the ongoing task" - how can specific task parameters be represented irrespective of the task? Please rephrase/fix logic.

Page 4: "Two male rhesus monkeys (Macaca mulatta) weighing between 6-8 kg underwent a unique surgery during" - unique in what sense - single surgery, or special / unusual surgery? If the latter, please specify the uniqueness.

"isomorphic scan" - you probably mean "isotropic"

Page 6

Baselinecorrected - typo

Page 7

"of two different macaque rhesus monkeys" - no need for "different"

Page 8

"While monkeys needed to enhance visual attention in the cued quadrant to respond to the target onset in the exogenous and endogenous attentional tasks (Amengual et al., 2022; Astrand et al., 2020, 2016), they needed to memorize the location of the cue to produce an oriented eye movement towards that location." - add "in the memory-guided saccade task" at the end.

Figure 2 legend: "full line" - "solid line".

Figure 2A,B: what is denoted by the gray shaded boxes?

Figure 4 legend: "Demixed principal components show similar explain variance as PCA." - typo, "explained".

Figure 4C - time axis labels are completely confused, there are three different variants (time to target, time from target, time from cue) and only the first one is correct, I believe.

Suppl. Figure 1: since it is MUA, I suggest to avoid usage of "Single Neuronal Activity" or "single neuron".

---

## [Decision Letter · Decision Letter 2]

16 Aug 2024

Dear Dr Ben Hamed,

Thank you for your patience while we considered your revised manuscript "Distinct neural states encode task identity in frontal eye field and interact with its core spatial properties" for publication as a Research Article at PLOS Biology. Your revised study has been evaluated by the PLOS Biology editors, the Academic Editor and the original reviewers.

In light of the reviews, which you will find at the end of this email, we would like to invite you to revise the work to thoroughly address the reviewers' reports.

As you will see below, Reviewer 1 is largely satisfied with the revision, but Reviewer 2 still has a lot of concerns regarding the analyses.

Given the extent of revision needed, we cannot make a decision about publication until we have seen the revised manuscript and your response to the reviewers' comments. Your revised manuscript is likely to be sent for further evaluation by all or a subset of the reviewers.

**IMPORTANT - SUBMITTING YOUR REVISION**

*Re-submission Checklist*

*Published Peer Review*

*PLOS Data Policy*

*Blot and Gel Data Policy*

Sincerely,

Christian

Christian Schnell, PhD

Senior Editor

PLOS Biology

cschnell@plos.org

REVIEWS:

Reviewer #1: The authors addressed most of the points I raised in a satisfactory manner. There is one point that is still not clear to me in the data preprocessing for multi unit activity as shown in Fig. 2A, B.

In Fig. 2A where an example MUA is shown it is now mentioned that "The data was z-scored relative to mean and standard deviation of average individual trial responses during baseline (-150 to 0 ms before cue onset)." I understand that this refers to the average across individual trials for this channel. However, the baseline level is around 2. I would have expected it to be around 0 if the mean baseline was subtracted. Am I missing something? Similarly, in Fig. 2B where the population average is shown it is around 0.2 instead of 0. Could you please explain why?

Reviewer #2: Dear authors, dear editor,

The manuscript has substantially improved; the revision clarified or fixed several issues. However, four major questions remain unresolved, and I believe they need to be carefully addressed to make the study convincing enough for publication in PLoS Biology. Additionally, there are still parts of the writing that are unclear or confusing. The manuscript also contains several stylistic inconsistencies and typos, so a thorough review of the text is recommended to ensure clarity and consistency.

Major point 1: Claims of mixed selectivity using MUA

I am afraid I am not fully convinced by the authors' response that "A recent publication indicates that dimensionality reduction methods are accurate in the absence of spike sorting (Trautmann et al., 2019).". The paper by Trautmann and colleagues shows that the neural population dynamics (neural space trajectories) can be estimated accurately enough without the spike sorting. As they write: "The use of multiunit threshold crossings is not appropriate for establishing whether neurons exhibit selective tuning to one or more stimuli or task parameters. … The experimental neuroscience paradigm proposed here, using threshold crossings, applies to neural population-level analyses. It is not applicable in cases where one wishes to make statements about the properties (e.g., stimulus selectivity) of individual neurons." (Trautmann et al., 2019). In the current manuscript, the authors write, e.g. in the abstract, that "The PFC cells show a non-linear mixed selectivity characterized by specific tuning for multiple task- and behavior related parameters.", so the study is about cells, not about population neural trajectories. As authors themselves state, based on the previous work that used the spike sorting: "Mixed selectivity corresponds to the ability of neurons to simultaneously encode different sources of information (Fusi et al., 2016; Rigotti et al., 2013)." All in all, mixed selectivity is the property of individual neurons. I think it goes without saying that the FEF "population response" would represent both the tasks (which are quite different in a number of features, including very important aspect of the eye vs hand effector for the FEF) as well as the spatial position effects. At least for me, the real question is if and how this information is multiplexed at the level of single neurons.

Furthermore, these dimensionality-reduction arguments do not apply at all to the ANOVA-based results, which are currently an integral part of the manuscript. Therefore, I think the authors should provide clear evidence for the mixed selectivity using spike-sorted data. I know it is a lot of work, so perhaps it can be done on a subset of single neurons with the best SNR. A consistent spike-sorting throughout the entire duration of a session (i.e. across all three tasks) will also allow addressing the major point 2 below.

Major point 2: Recording (in)stability, fixed thresholding and blocked task design

The authors applied a constant (and arguably, very lenient) 3 times SD threshold to derive MUA spike trains from the filtered data. My concern was that this could lead to different firing rates, and different composition of MUA, due to temporal drifts in the acute linear probe recordings. I take the authors' point that the dPCA can deal with the drifts by assigning the related variance to non-specific components. But the task and the time are not orthogonal in the current design, so for instance the strongest task-related component 3 in Figure 4 could be related to time, as the new analysis in Supplementary material M2 also seems to show. Furthermore, the ANOVA analyses are not at all immune to drifts. Therefore, I think illustrative examples of filtered recordings as a function of time (from which the spike crossings were derived) should be shown, and most importantly, the analysis of raw firing rates as a function of time and the task block should be presented, to dissociate the effects of the tasks and potential drift effects. Again, using the well-isolated single neuron data can also help to address this concern, if the spike sorting is stable and accurate enough despite potential amplitude drifts.

"A threshold was applied independently for each recording contact and before the actual task-related recordings started, such that the crossing of this threshold resulted in multiunit spike trains. This threshold corresponded to 3 standard deviations from mean activity during unconstrained eye movements by the monkeys while exploring a blank screen. This threshold was confirmed during the offline analyses." - What does "confirmed" mean here? Was the threshold ever adjusted offline? If not, how much it deviated from the initially calculated/guessed 3 SD?

Major point 3: Interpretation of ANOVA results and mixed selectivity, relationship to dPCA results

One of the central points the authors advance in this work is the non-linear mixed selectivity. If I understand correctly, this type of mixed selectivity should be reflected in the interaction term of the two-way ANOVA. But the analysis in the Figure 2D combines the units which show an additive combination of the two main effects (presumably corresponding to linear mixed selectivity) and those showing the interaction. Please separate these two effects, which would also allow to relate this to the dPCA interaction component.

Additionally, please describe in simple terms how the (non-linear) mixed selectivity influences the preferred spatial position in individual neurons/multiunits? Specifically, please describe whether and how the actual preferred position changes as a function of the task: e.g., as shifts of the preferred position, or changes in the "depth" of spatial tuning (e.g. gain-like task effect) but without the shifts of the preferred position? It would be helpful to show the proportions of units where such changes occur, along with illustrative examples of tuning curves (across all four positions) for both linear and non-linear mixed selectivity units. As a general note, the Results section is very condensed; I suggest taking advantage of the PLoS Biol format to include more than just four figures to better illustrate the findings.

As a control, and again to better relate the results of the ANOVA analysis to subsequent dPCA not only for the post-cue epochs but also for pre-cue epoch, please repeat the same two-way ANOVA analysis on the pre-cue epoch. The expectation would be that the "non-causal" position selectivity should largely disappear, except for some spurious random effects due to false positives.

Finally, in the current manuscript the authors use the toolbox by Kobak et al. to generate Figures 3 and 4, and do not go any deeper beyond the basic output. But the dimensions underlying main effects (tasks and position) as well as the non-linear mixed selectivity can be distributed very differently across the units. For instance, if one imagines a 2D grid plot where one axis is units and another dimensions, and the intensity of a cell indicates the "weights" for each unit and each dimension, there could be for instance two different scenarios that both would result in representation of both types of information at the population level. In the first scenario, one group of units can represent e.g. the position dimensions but not the task dimensions, and the complementary group - vice versa, whereas in the second scenario, all units would have some non-zero weights for both types of dimensions. If I understand correctly, the current analysis cannot distinguish between these two scenarios? Did the authors examine the patterns of the weight distributions in their analysis?

Major point 4: There are a number of inconsistencies / seemingly arbitrary choices in respect to many variants of baseline correction, selected analysis epochs and ANOVA vs dPCA results.

Figure 2: "The data was z-scored relative to mean and standard deviation of average individual trial responses during baseline (-150 to 0 ms before cue onset)." - what does "average individual" mean? Why the normalized pre-cue activity level is not around zero given that largely overlapping window -150 to 0 ms was used for the baseline? Same questions about population response in B. I also don't understand why B is not just an average of already normalized responses of all MUA as in A - i.e. why there seems to be different normalization applied?

Next, I don't understand why the new Supplementary Figure 2, related to Figure 2, shows a different scale for the "normalized firing rate" - should it not be the same as in the Figure 2A? I also assume that this is not the same normalization that is used for dPCA (see below), because subtracting a baseline should result in in firing rate difference to baseline in Hz, not in very small a.u.?

Finally, there is yet another normalization for dPCA: "To extract the averaged firing rates for dPCA analysis, these epoched MUA data were baseline corrected (independently for each channel and each trial) relative to the time period from -500 ms to -300 ms before the cue. This reduced trial-to-trial variability in baseline activity levels and reduces individual trial differences." The question is why a different baseline epoch has been used for dPCA?

"First, MUA activity previous to the cue onset was epoched from -500 ms to 0 with respect to the onset of the Cue for the Pre-Cue analysis." - why to normalize per trial using the window (-500 ms to -300 ms before the cue) that is largely overlapping with the epoch of interest? Do I understand correctly that if you were to normalize in -500 ms to 0 ms before the cue, you would not be able to see pre-cue effects at all?

Alternatively, there is an error in the description above, because the Results text (page 12) says "This analysis was firstly performed during the precue period (200 ms before cue onset to cue onset) and will be later performed during the cue to target interval (500 ms to 0 ms before target onset) to unmix task- from sensory- specific variance." Is the "precue period (200 ms before cue onset to cue onset)" supposed to be the same as "Pre-Cue" epoch above?

Figure 2: "Gray box corresponds to the delay period during which neuronal activities were analyzed for subplots C and D." - so it is 500 ms for the two-way ANOVA" - so it is 500 ms epoch for two-way ANOVA, on raw firing rates or on normalized (and so, how) firing rates? But for the "Wilcoxon ranksum test was performed between baseline and time interval for each channel and position across trials." - here the "time interval" (please provide better description) is, according to the Methods, is different, 800 - 900 ms after the cue onset: "First, a non-parametric Wilcoxon test was used to identify responsive channels, i.e. channels for which the activities observed during the baseline and during the post cue period (800 to 900 ms following cue onset, only considering trials for which target or fixation off event happened 1000ms after cue presentation) were significantly different for at least one of the four possible spatial cued positions (p<0.05)."

Minor

The Introduction and Discussion sections contain some areas that remain unclear (please see specific points below). Additionally, the Introduction does not adequately articulate the gap in the literature that the authors aim to address. The fact that the PFC, particularly the FEF, is known to represent a variety of variables and contingencies is well-established. Therefore, what makes this study's findings novel? What does this research contribute to the already extensive body of knowledge? I also suggest addressing a contrasting perspective on mixed coding in the FEF within the Discussion (cf. reference below), to provide a balanced counterpoint.

Kaleb A Lowe, Wolf Zinke, Joshua D Cosman, Jeffrey D Schall, Frontal eye fields in macaque monkeys: prefrontal and premotor contributions to visually guided saccades, Cerebral Cortex, Volume 32, Issue 22, 15 November 2022, Pages 5083-5107, https://doi.org/10.1093/cercor/bhab533

Page 3

"Each of these tasks recruited specific cognitive functions: endogenous attention, exogenous attention and spatial working memory." - as has been argued previously (and agreed upon by the authors), the spatial working memory is present in all 3 tasks, the main differences beyond attentional demands are the presence of visual stimuli placeholders during the delay in two attention tasks, and the manual response vs saccades. I think it needs to be mentioned already in the Introduction.

"While monkeys were performing these tasks in independent blocks or trials," - blocks ***of*** trials

"…, dense recording probes were used to record multi-unit activity (MUA)…" - 24-contact Plexon U-probes with 250 micron distance between sites are not really dense recording probes (unlike e.g. Neuropixels where contacts are very close and each neuron can be seen on several channels - here the channels are independent).

"This result suggests that task identity and spatial information are not encoded independently in absolute terms such that the specific state of one variable influences the specific computations implemented by the other variable." - I cannot fully understand this sentence, in particular the "such that" logical transition - do you mean that two types of information are ***not encoded independently*** so that one variable does influence the other?

Note that the next (last) paragraph of the Introduction seems to claim the opposite: "In addition, spatial information is represented ***independently*** from task-related information, indicating that task-identity is represented at the same hierarchical level as specific task parameters such as position.

Page 4

"At the fundamental level, it will be important to pursue on this work in order to better understand the specific dimensions that the prefrontal cortex uses to represent task identity and achieve perceptual representation stability across tasks. At the translational level, this work opens new venues for the implementation of brain computer interface technologies based on machine learning algorithms able to extract complex cognitive information from realtime PFC recordings." - but in the next paragraph, this statement is again repeated:

"At the fundamental level, it will be important to pursue on this work in order to better understand the specific dimensions that the prefrontal cortex uses to represent task identity. At the translational level, this work opens new venues for the implementation of brain computer interface technologies based on machine learning algorithms able to extract complex cognitive information from real-time PFC recordings."

Page 16

"These three tasks presented a very similar structure (the onset of a cue was followed by a target stimuli)." - and spatially-contingent manual or saccade response?

"The memory guided saccade task mostly recruited spatial working memory as well as spatially

accurate sensorimotor transformation processes." - The attention tasks also required sensorimotor transformation: visual target detection - manual response. Please amend this statement.

"We found that, at the neural level, the dynamics of the activity of the FEF cells was cued was different across tasks, suggesting a different encoding of the sensory information as a function of the task being performed." - "was cued was different" does not work. More importantly, I am not at all sure how is the dynamics addressed in this study?

"We found that the FEF encoded task identity in two sources of task-related variability

(independently from the processing of sensory information) that segregated in orthogonal components that conformed a two-dimensional space, …" - "that segregated … that conformed" does not make sense.

"…high dimensional representation of the neuronal population" - no, as the authors wrote in the paragraph above, it is "high dimensional representation of sensory information".

"Source data for the figures and associated code can be downloaded at the following link:"

https://osf.io/z8eh9/?view_only=a445fa5522924f709cca985b97ae7e9e

-- It is empty?

"Supplementary material M2: … In contrast with other methods such as decoding or PCA, the analysis of neuronal variability that we implemented in this work relies on demixed principal component analysis (dPCA, see below). This method has been sho" - missing the end of the sentence?

---

## [Decision Letter · Decision Letter 3]

22 May 2025

Dear Suliann,

Thank you for your patience while we considered your revised manuscript "Distinct neural states encode task identity in frontal eye field and interact with its core spatial properties" for consideration as a Research Article at PLOS Biology. Your revised study has now been evaluated by the PLOS Biology editors, the Academic Editor and one of the original reviewers.

In light of the reviews, which you will find at the end of this email, we are pleased to offer you the opportunity to address the remaining points from the reviewers in a revision that we anticipate should not take you very long. We will then assess your revised manuscript and your response to the reviewers' comments with our Academic Editor aiming to avoid further rounds of peer-review, although we might need to consult with the reviewers, depending on the nature of the revisions.

**IMPORTANT - SUBMITTING YOUR REVISION**

*Resubmission Checklist*

*Published Peer Review*

*PLOS Data Policy*

*Blot and Gel Data Policy*

Sincerely,

Christian

Christian Schnell, PhD

Senior Editor

PLOS Biology

cschnell@plos.org

REVIEWS:

Reviewer #2: The manuscript has improved substantially; most of my points have been addressed and necessary corrections have been made. In particular, the addition of the single unit data, and the updated reporting of ANOVAs, made the results more convincing.

I have few suggestions/questions for the final version prior to the publication:

1. Please systematically summarize your data sample: how many sessions (12 per monkey based on the behavioral data?), how many valid channels, and most importantly, how many sessions/channels have been successfully spike-sorted and how many single units have been extracted and analyzed.

2. My sincere apologies, but even after all error fixes and explanations, I still cannot follow the logic of different baselines: -500 to -300 before the cue onset vs -500 to 0 before the cue onset (e.g. both used in the Figure 2)?

What is the pre-cue epoch: "pre-cue epoch (-500 to 0ms prior to cue presentation, supplementary Figure 5B)" -- page 12, or "pre-cue period (200 ms before cue onset to cue onset)" -- page 14?

Then there is also another period "averaged in the time interval -300 to - 100ms pre Cue" in the Suppl. Figures 8 and 9...

3. It would be good to see the firing rates of MUA and SUA as a function of time in the session.

4. Concerning the new, very important for illustrations of mixed selectivity, Figure 3 and the associated text: it would be interesting to see the examples of linear and non-linear (i.e. with interaction) mixed selectivity MUA, and also corresponding examples of SUA. The population averages of MUA in 3C and D do not make a convincing case as they do not look that much different from each other.

"Understanding why these shifts seem to be specific to mixed-selectivity MUAs with interaction requires further investigations." - I am not sure if I am following this statement. Shouldn't that be the case by definition? If there is an RF shift, doesn't the unit have to show mixed selectivity?

5. Lastly, I still find the expression "single MUA" confusing and would suggest using "an example MUA" in the corresponding places.

---

## [Editor Report · Decision Letter 4]

22 Jul 2025

Dear Suliann,

Thank you for your patience while we considered your revised manuscript "Distinct neural states encode task identity in frontal eye field and interact with its core spatial properties" for publication as a Research Article at PLOS Biology. This revised version of your manuscript has been evaluated by the PLOS Biology editors and the Academic Editor.

Based on our Academic Editor's assessment of your revision, we are likely to accept this manuscript for publication, provided you satisfactorily address the following data and other policy-related requests:

* We would like to suggest a different title to improve its accessibility for our broad audience: "The prefrontal cortex encodes task-identity information and flexibly adjusts its sensory processes as a function of the specific ongoing task"

* Please add the links to the funding agencies in the Financial Disclosure statement in the manuscript details.

* DATA POLICY:

Regardless of the method selected, please ensure that you provide the individual numerical values that underlie the summary data displayed in the following figure panels as they are essential for readers to assess your analysis and to reproduce it: 1B, 4DE, M1, M2 and M13ABC

* CODE POLICY

We expect to receive your revised manuscript within two weeks.

*Published Peer Review History*

*Press*

Sincerely,

Christian

Christian Schnell, PhD

Senior Editor

cschnell@plos.org

PLOS Biology

---

## [Editor Report · Decision Letter 5]

6 Aug 2025

Dear Suliann,

Thank you for the submission of your revised Research Article "The prefrontal cortex encodes task-identity information and flexibly adjusts its sensory processes as a function of the specific ongoing task" for publication in PLOS Biology. On behalf of my colleagues and the Academic Editor, Raphael Kaplan, I am pleased to say that we can in principle accept your manuscript for publication, provided you address any remaining formatting and reporting issues. These will be detailed in an email you should receive within 2-3 business days from our colleagues in the journal operations team; no action is required from you until then. Please note that we will not be able to formally accept your manuscript and schedule it for publication until you have completed any requested changes.

PRESS

Sincerely, 

Christian

Christian Schnell, PhD

Senior Editor

PLOS Biology

cschnell@plos.org